# SRT: Super-Resolution for Time Series via Disentangled Rectified Flow

**Jufang Duan, Shenglong Xiao & Yuren Zhang**
Bytedance
{duanjufang,xiaoshenglong,zhangyuren}@bytedance.com

## Abstract

Fine-grained time series data with high temporal resolution is critical for accurate analytics across a wide range of applications. However, the acquisition of such data is often limited by cost and feasibility. This problem can be tackled by reconstructing high-resolution signals from low-resolution inputs based on specific priors, known as super-resolution. While extensively studied in computer vision, directly transferring image super-resolution techniques to time series is not trivial. To address this challenge at a fundamental level, we propose **S**uper-**R**esolution for **T**ime series (SRT), a novel framework that reconstructs temporal patterns lost in low-resolution inputs via disentangled rectified flow. SRT decomposes the input into trend and seasonal components, aligns them to the target resolution using an implicit neural representation, and leverages a novel cross-resolution attention mechanism to guide the generation of high-resolution details. We further introduce SRT-large, a scaled-up version with extensive pre-training, which enables strong zero-shot super-resolution capability. Extensive experiments on nine public datasets demonstrate that SRT and SRT-large consistently outperform existing methods across multiple scale factors, showing both robust performance and the effectiveness of each component in our architecture.

## 1 Introduction

The availability of fine-grained, high-resolution time series data is critical for the accuracy and effectiveness of downstream analytics and decision-making across numerous domains. In healthcare, for instance, high-resolution electrocardiogram signals are essential for detecting subtle but clinically critical arrhythmias that are often obscured in lower-frequency recordings (Kachuee et al., 2018; Hannun et al., 2019). Similarly, in industrial IoT, vibration data sampled at kilohertz rates significantly improves the precision of machinery prognostics and early fault detection (Zhao et al., 2017; Lei et al., 2020). The field of climatology also heavily relies on temporally dense data to model complex weather phenomena and extreme events (Sillmann et al., 2017; David et al., 2022). However, the acquisition of such high-resolution data is often impeded by domain-specific constraints, including limited device battery life, communication bandwidth, storage costs, and computational overhead (Dai et al., 2020). These limitations collectively make the continuous collection of high-resolution data economically infeasible or physically impossible, resulting in the widespread prevalence of coarsely sampled or aggregated time series.

To address this issue, we aim to develop a method that generates high-resolution data from existing low-resolution inputs. Such methods have been extensively studied in the computer vision (CV) field under the name of image super-resolution, which leverages techniques such as generative adversarial networks (Ledig et al., 2017), diffusion models (Li et al., 2022), and flow matching (Lugmayr et al., 2020) to synthesize visually realistic high-resolution images from their low-resolution counterparts. Analogous to image super-resolution, we seek to devise a time series super-resolution (TSSR) method that generates values between consecutive observed points, thereby mapping low-temporal-resolution time series to high-temporal-resolution ones. Although the adaptation of image super-resolution techniques to time series is conceptually appealing, there are fundamental challenges due to the intrinsic differences between the two data types. Firstly, the priors governing natural images and those required for time series are fundamentally different. Secondly, the data dimensionality varies significantly, and the axes subject to upscaling are not the same. These differ-

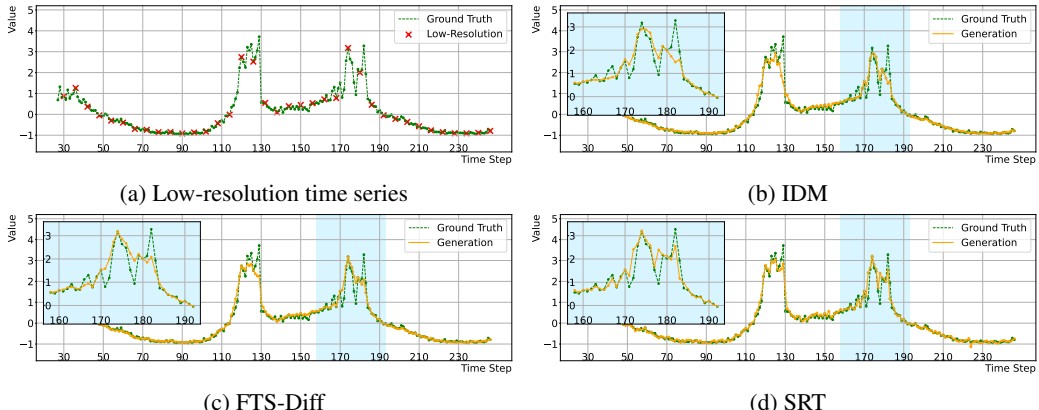

Figure 1: Qualitative results on a segment from traffic domain. Compared to the leading approaches from image super-resolution and time series generative models, our SRT more faithfully reconstructs the overall profile of the high-resolution ground truth in both stable and volatile phases.

ences pose unique challenges for the direct application of image super-resolution approaches, whose performance, as a result, often falls short. On the other hand, the task of generating missing values in time series has been extensively studied within the imputation literature (Tashiro et al., 2021; Yıldız et al., 2022; Duan et al., 2024). These works share a common high-level goal with TSSR, *i.e.*, to infer plausible data points based on observed context. However, a critical distinction lies in the nature of the missingness. Imputation typically handles with arbitrarily missing points within an originally high-resolution sequence, whereas TSSR aims to synthesize a fundamentally high-resolution signal from a systematically downsampled input. Consequently, while imputation can often rely on assumptions like local smoothness or global consistency, TSSR must generate credible high-resolution components, such as sharp peaks or transient vibrations, which are absent in low-resolution input. This distinction renders TSSR a more challenging and under-explored problem, which calls for more powerful generative models and more informed priors to guide the synthesis of plausible high-resolution details.

Building upon the aforementioned challenges, we begin by formally distinguishing two fundamental types of TSSR problems based on the genesis of the low-resolution data, *i.e.*, sampled super-resolution (SSR) and aggregated super-resolution (ASR), whose core challenge differs critically. SSR must reconstruct missing samples from an undersampled sequence, while ASR must distribute an aggregated value into its fine-grained constituents. This makes ASR inherently more ambiguous and ill-posed, as the original high-frequency distribution is entirely lost, leaving only a statistical summary. To address these dual challenges within a unified framework, we propose **S**uper-**R**esolution for **T**ime series (SRT), whose key idea is to disentangle the rectified flow-based super-resolution process via time series decomposition and to guide the generation of high-resolution details with informative cues derived from the low-resolution series. Specifically, our SRT framework operates through a structured pipeline. First, the input low-resolution series is decomposed into its trend and periodic components. These components are then temporally aligned to the target scale using an implicit time function (ITF), which employs a continuous implicit neural representation to act as a versatile and learnable interpolator. Subsequently, two separate rectified flow models are employed to generate the residual high-resolution details, with one for trend and the other for periodic component. This dual-path design not only captures distinct temporal dynamics but also enhances interpretability by isolating the contributions of the two components to the final output. To effectively fuse the temporally aligned condition from the ITF and predict the velocity field governing the state transitions, we introduce a novel cross-resolution attention (CRA) mechanism within a decoder-only velocity predictor. By integrating decomposition, continuous alignment, and conditioned generation, SRT effectively constrains the solution space, enabling high-fidelity reconstruction for both SSR and the more challenging ASR tasks. Extensive experiments demonstrate that SRT not only outperforms the baselines in both pointwise and overall accuracy, but also achieves superior reconstruction of high-resolution details. Furthermore, we validate the effectiveness of each component through ablation studies, and our experimental results confirm that the proposed velocity

predictor significantly enhances overall model performance. A preview of our results is visualized in Figure. 1.

In addition to the standard SRT introduced above, we further extend the model to address the challenge of inaccessible high-resolution time series in certain TSSR scenarios. Specifically, we propose SRT-large, which achieves zero-shot super-resolution capability. Compared to the standard version, SRT-large significantly increases the parameter size and is pretrained on large-scale datasets across various domains, enabling the model to generalize to previously unseen types of time series and perform super-resolution without requiring high-resolution training samples. Experimental evidence demonstrate that, even in zero-shot setting, SRT-large achieves state-of-the-art performance on diverse datasets. Moreover, it provides more consistent results than baselines at different scale factors.

Our main contributions can be summarized as the following four parts.

(1) We formally define two subtypes of time series super-resolution (TSSR) problems. Furthermore, we propose a standardized evaluation protocol and experimental workflow for assessing performance on TSSR tasks, which can serve as a benchmark for future research in this area.

(2) We introduce the SRT model, which leverages time series decomposition and rectified flow. The framework incorporates an implicit time function (ITF) for condition generation and a velocity predictor for multi-resolution feature fusion between coarse-grained and fine-grained time series.

(3) We extend the standard SRT to SRT-large by increasing model capacity and conducting large-scale pretraining across diverse domains, thereby equipping it with zero-shot TSSR capabilities.

(4) Extensive experiments demonstrate the effectiveness of our proposed method. Each core component, including ITF, CRA, and disentanglement, contributes positively to the overall performance.

## 2 RELATED WORK

Super-resolution (SR) has been a fundamental problem in computer vision (CV), aiming to reconstruct high-resolution images from low-resolution counterparts. Methods have experienced rapid progress in recent years, moving beyond traditional convolutional (Dong et al., 2014) and GAN-based methods (Ledig et al., 2017; Wang et al., 2018) toward more powerful generative models. Diffusion models, including SR3 (Saharia et al., 2022), CDM (Ho et al., 2022), LDMs (Rombach et al., 2022), and SRDiff (Li et al., 2022), leverage iterative denoising for high-quality reconstruction. Flow matching and its variants further improve sample efficiency and quality by aligning the generative process with the data distribution (Lugmayr et al., 2020; Liang et al., 2021). The rectified flow framework, in particular, offers improved convergence and sample quality, providing new perspectives for SR research (Zhu et al., 2024). Although the above methods have shown great effectiveness, directly transferring these approaches from CV to time series often yields unsatisfactory results, mainly due to the gap between required priors as well as the differences in signal space.

In the time series domain, most studies have focused on imputation rather than super-resolution. Imputation has long been considered a hot research topic and been enrolled as a downstream task to verify the effectiveness of various models including universal time series models (Wu et al., 2022; Wang et al., 2024), contrastive learning (Liu & Chen, 2024; Duan et al., 2024), and diffusion models (Tashiro et al., 2021; Alcaraz & Strodthoff, 2022). Other works leverage the power of imputation for tasks like anomaly detection (Chen et al., 2023; Xiao et al., 2023) or representation learning (Senane et al., 2024). However, imputation and super-resolution are fundamentally different *w.r.t.* available data volume, target sampling rate, and explicitly available information (see more in Appendix. B). As a result, directly using imputation methods to solve TSSR task may not realize plausible outcomes.

Another relative category of methods is time series generative models. Apart from the well-studied VAE (Desai et al., 2021; Li et al., 2023) and GAN (Smith & Smith, 2020; Jeon et al., 2022), models based on diffusion and flow matching remain an active area of investigation. Yuan & Qiao (2024) integrates time series decomposition with diffusion modeling, resulting in interpretable and high-fidelity time series generation. Huang et al. (2024) performs time series segmentation and clustering, and trains diffusion models on representative segments before generating entire sequences recursively through a Markov chain approach. Zhang et al. (2024) and Tamir et al. (2024) utilize flow matching for time series generation, avoiding problems like slow sampling speed, inconsistency between training and inference, and noise accumulation that are commonly observed in diffusion

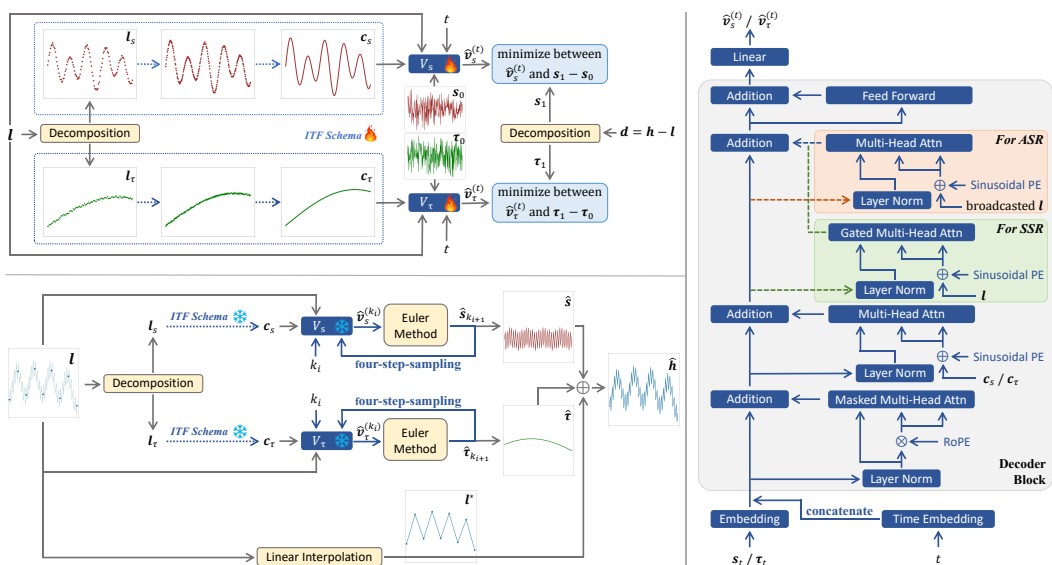

Figure 2: Architecture of our proposed SRT. **The upper left** shows the training process, where the true residual sequence is decomposed, and the velocity predictors ($V_s$ and $V_\tau$) are trained to fit the difference between the true values of $s$ and $\tau$ and their respective initial states. **The lower left** depicts the inference process. The predictions $\hat{s}$ and $\hat{\tau}$ are obtained using predicted velocity via the Euler method. Summing these predictions yields the estimated residual sequence, which is then added to the linear interpolated low-resolution input to produce the final TSSR result. **The right side** presents the structure of the proposed velocity predictor, which adopts a decoder-only architecture and incorporates a specially designed cross-resolution attention mechanism for velocity prediction, conditioning on both the low-resolution input and features aligned by the ITF.

models. These types of models may address the TSSR problem by conditioning on low-resolution time series to generate high-resolution outputs. However, without explicitly modeling the LR-to-HR correspondence, the generated high-resolution series may lack structural consistency and detailed fidelity, which is particularly critical in TSSR scenarios.

# 3 PROPOSED METHOD

## 3.1 PRELIMINARY

Given a set of low-resolution time series $\mathbb{L} = \{l_i | i = 1, 2, ..., N\}$, where $l_i \in \mathbb{R}^{L \times D}$, $L$ is the length and $D$ is the channel size, the goal of SRT is to generate the high-resolution time series $\mathbb{H} = \{h_i | i \in 1, 2, ..., N\}$, where $h_i \in \mathbb{R}^{H \times D}$ and $H$ is the length of the target high-resolution series. The scale factor is defined as $\alpha = \left\lfloor \frac{H-1}{L-1} \right\rfloor$, with $\alpha - 1$ equals to the number of data points to be generated between consecutive low-resolution time steps. We use $l_i^{(t)}$ and $h_i^{(t)}$ to represent the $t$-th time step of $l_i$ and $h_i$, respectively. Two univariate sequence are utilized to depict the correspondence between $l_i$ and $h_i$, which are target mask sequence $m = \{m_j | j = 1, 2, ..., H\}$ and mask index sequence $p = \{p_k | k = 1, 2, ..., L\}$. We set $m_j = 1$ if the $j$-th time step of the $h_i$ corresponds to a given low-resolution value and $m_i = 0$ if the time step should be generated. And $p = \{j | m_j = 1, \ i = 1, 2, ..., H\}$ is used to represent the index of the given low-resolution data points.

By examining a wide range of real-world requirements, we categorize TSSR into two cases, namely the sampled super-resolution (SSR) and the aggregated super-resolution (ASR).

***Definition 1:*** For SSR, each $l_i$ is subject to $l_i^{(k)} = h_i^{(p_k)}$

***Definition 2:*** For ASR, each $l_i$ is subject to $l_i^{(k)} = \frac{1}{\alpha} \sum_{j=p_k}^{p_{k+1}} h_i^{(j)}$

In practice, the SSR is encountered in scenarios like generating high frequency signals from low sampling rate sensors, while the ASR can be used to decompose the statistical value with a temporal window into finer granularity, *e.g.*, disaggregating daily average precipitation into hourly one. SRT can handle both of the two super-resolution tasks.

## 3.2 ARCHITECTURE

SRT tackles TSSR by generating the residual between $\boldsymbol{h}_i$ and the interpolated low-resolution series $\boldsymbol{l}_i^*$, that is, reconstructing the detail sequence, namely $\boldsymbol{d}_i$, lost by the coarse-grained time series compared to its fine-grained counterpart. We employ the decomposition method of Autoformer (Wu et al., 2021) to disentangle the detail sequence into trend component $\boldsymbol{\tau}_i$ and periodic component $\boldsymbol{s}_i$:

$$\boldsymbol{d}_i = \boldsymbol{s}_i + \boldsymbol{\tau}_i, \quad \text{with } \boldsymbol{\tau}_i = \text{AvgPool}(\text{Padding}(\boldsymbol{d}_i)), \tag{1}$$

and then model each component in parallel using two separate processes. This approach not only facilitates model fitting but also enhances the interpretability of the entire model by disentangling the generation of different temporal dynamics from one another. For example, in the scenario where daily precipitation is disaggregated into hourly one, the generation of trend components reflects the moving tendency of high-resolution precipitation details, while the generation of periodic components reveals the short-term regular fluctuations. See Appendix. G.3 for thorough analysis.

The generation of $\boldsymbol{s}$ and $\boldsymbol{\tau}$ [1] is accomplished by rectified flow (Liu et al., 2022), who learns the transformation from the prior distribution $\pi_0$ to the target distribution $\pi_1$ by solving an ODE. Specifically, let $\boldsymbol{s}_0 \sim \pi_0^{(s)}$ and $\boldsymbol{s}_1 \sim \pi_1^{(s)}$ represent the start and target state of $\boldsymbol{s}$, and $\boldsymbol{\tau}_0 \sim \pi_0^{(\tau)}$ and $\boldsymbol{\tau}_1 \sim \pi_1^{(\tau)}$ represent the start and target state of $\boldsymbol{\tau}$, the velocity of the transportation can be formulated as

$$\boldsymbol{v}_s(\boldsymbol{s}_t, t) = \frac{d\boldsymbol{s}_t}{dt}, \quad \boldsymbol{v}_t(\boldsymbol{\tau}_t, t) = \frac{d\boldsymbol{\tau}_t}{dt}, \quad t \in [0, 1],$$

where $\boldsymbol{s}_t$ and $\boldsymbol{\tau}_t$ is the state at time $t$ for the seasonal and trend component, respectively. Given a path where both $\boldsymbol{s}_t$ and $\boldsymbol{\tau}_t$ can be defined as the linear interpolation between the start and the target state, *i.e.*, $\boldsymbol{s}_t = t\boldsymbol{s}_1 + (1-t)\boldsymbol{s}_0$ and $\boldsymbol{\tau}_t = t\boldsymbol{\tau}_1 + (1-t)\boldsymbol{\tau}_0$, and two structurally identical velocity predictors $V_s(\,\cdot\,; \boldsymbol{\theta}_s)$ and $V_\tau(\,\cdot\,; \boldsymbol{\theta}_\tau)$ for the prediction of $\boldsymbol{v}_s$ and $\boldsymbol{v}_\tau$, the parameter $\boldsymbol{\theta}_s$ and $\boldsymbol{\theta}_\tau$ defines the two predictors can be simultaneously optimized by

$$\min \int_0^1 \mathbb{E}\left[(\boldsymbol{s}_1 - \boldsymbol{s}_0 - V_s(\boldsymbol{s}_t, t, \boldsymbol{c}_s, \boldsymbol{l}; \boldsymbol{\theta}_s))^2 + (\boldsymbol{\tau}_1 - \boldsymbol{\tau}_0 - V_\tau(\boldsymbol{\tau}_t, t, \boldsymbol{c}_\tau, \boldsymbol{l}; \boldsymbol{\theta}_\tau))^2\right] dt,$$

where $\mathbb{E}[\cdot]$ represents the expected value, $\boldsymbol{c}_s$ and $\boldsymbol{c}_\tau$ are the conditions derived from ITF.

The rectified flow enables fast sampling while maintain the high fidelity generation result. When generation, we use a four-step-sampling and each step is based on the Euler method as

$$\hat{\boldsymbol{s}}_{k_{i+1}} = \hat{\boldsymbol{s}}_{k_i} + (k_{i+1} - k_i)V_s(\hat{\boldsymbol{s}}_{k_i}, k_i, \boldsymbol{c}_s, \boldsymbol{l}; \boldsymbol{\theta}_s), \quad \hat{\boldsymbol{\tau}}_{k_{i+1}} = \hat{\boldsymbol{\tau}}_{k_i} + (k_{i+1} - k_i)V_\tau(\hat{\boldsymbol{\tau}}_{k_i}, k_i, \boldsymbol{c}_\tau, \boldsymbol{l}; \boldsymbol{\theta}_\tau),$$

where $k_i$ is the sampling step, and the prediction is initialized as $\hat{\boldsymbol{s}}_0 \sim \mathcal{N}(0, I)$ and $\hat{\boldsymbol{\tau}}_0 \sim \mathcal{N}(0, I)$.

After the four-step linear transition to the target state, the final results can be gained by

$$\hat{\boldsymbol{h}} = \boldsymbol{l}^* + \hat{\boldsymbol{s}} + \hat{\boldsymbol{\tau}}, \quad \text{with } \hat{\boldsymbol{s}} = \hat{\boldsymbol{s}}_4 \text{ and } \hat{\boldsymbol{\tau}} = \hat{\boldsymbol{\tau}}_4.$$

We summarize the aforementioned workflow in Figure 2.

## 3.3 IMPLICIT TIME FUNCTION

Inspired by the success of implicit neural representation in related work (Chen et al., 2021; Gao et al., 2023), we come up with the implicit time function (ITF) capable of aligning the granularity gap on time axis. In SRT, the ITF takes low-resolution periodicity $\boldsymbol{l}_s \in \mathbb{R}^{L \times D}$ and low-resolution trend $\boldsymbol{l}_\tau \in \mathbb{R}^{L \times D}$ as inputs and generates high-resolution conditions. Similar to Eq. 1, the decomposition of $\boldsymbol{l}$ can be formulated as

$$\boldsymbol{l} = \boldsymbol{l}_s + \boldsymbol{l}_\tau, \quad \text{with } \boldsymbol{l}_\tau = \text{AvgPool}(\text{Padding}(\boldsymbol{l})).$$

---

[1]For simplicity, we omit the subscript $i$ in the following text.

The outputs are $\boldsymbol{h}_s \in \mathbb{R}^{H' \times D}$ and $\boldsymbol{h}_\tau \in \mathbb{R}^{H' \times D}$, with the length being aligned from $L$ to $H'$. Note that this process involves two distinct time axes: the original time axis with length $L$, and the target time axis with length $H'$. To clearly distinguish between the two axes, we use superscripts $j^{(o)}$ and $j^{(t)}$ to denote the $j$-th time step on the original and on the target time axis, respectively. Besides, a function $T(\cdot)$ is utilized to describe the mapping between the coordinates of the same point on the two axes, *i.e.*, $T(j^{(o)}) = j^{(t)}$.

ITF is composed of three stages, *i.e.*, temporal enrichment, value prediction, and pattern smoothness. During temporal enrichment, all the channels within a dilated window are concatenated to enhance the contextual information available at each time step. Because of the long-term temporal dependencies present in time series data (Zhou et al., 2021), the use of a dilated window allows for an enlarged receptive field in the ITF, thereby capturing more comprehensive temporal patterns. With a hyperparameter $r$ defining the radius, temporal enrichment for $\boldsymbol{l}_s$ can be formulated as

$$\tilde{\boldsymbol{l}}_s^{j^{(o)}} = \text{Concat}\left(\left\{\text{Padding}(\boldsymbol{l}_s^{(j+\delta)^{(o)}})\right\}_{\delta \in \{\pm 2^i | i=0,1,\dots,r\} \cup \{0\}}\right), \tag{2}$$

Since the structure is symmetric *w.r.t.* $\boldsymbol{l}_s$ and $\boldsymbol{l}_\tau$, the temporal enrichment for $\boldsymbol{l}_\tau$ are analogous, with only the subscript $s$ being replaced by $\tau$ in Eq. 2.

Next, each value on the target time axis can be initially estimated in the value prediction stage. For the periodic component, given a candidate time step $j_c^{(o)}$ and its corresponding enriched value $\tilde{\boldsymbol{l}}_s^{j_c^{(o)}}$, the value of the $k$-th step on the target time axis can be predicted via a neural network $g(\,\cdot\,;\,\phi)$ parameterized by $\phi$, as

$$\hat{\boldsymbol{h}}_s^{k^{(t)}}[j_c^{(o)}] = g\left(\tilde{\boldsymbol{l}}_s^{j_c^{(o)}},\ k^{(t)} - T(j_c^{(o)});\ \phi\right). \tag{3}$$

Based on the same reason, the value prediction for the trend component can be rendered by replacing the subscript from $s$ to $\tau$ in Eq. 3.

Finally, we conduct pattern smoothness aiming at determining the candidate time steps and aggregating the preliminary estimations. For the trend component, the smoothness is impulsed to locality, where significant deviation from consecutive time steps should be avoid. Denote $j_n^{(o)}$ to be the nearest time step around $k^{(t)}$ on the original time axis satisfying $j_n^{(o)} = \text{argmin}_{j^{(o)}} |T(j^{(o)}) - k^{(t)}|$, and $w_{\Delta j} = \frac{1}{|k^{(t)} - T((j_n + \Delta j)^{(o)})|}$ to be the weight, the pattern smoothness can be defined as

$$\tilde{\boldsymbol{h}}_s^{k^{(t)}} = \sum_{\Delta j \in [-1,\ 1]} \frac{w_{\Delta j}}{w} \hat{\boldsymbol{h}}_s^{k^{(t)}}\left[(j_n + \Delta j)^{(o)}\right], \quad \text{with} \quad w = \sum_{\Delta j \in [-1,\ 1]} w_{\Delta j}. \tag{4}$$

For periodic input series, however, the recursively appeared pattern should also be taken into consideration,which means the smoothness should not only be constraint to locality, but to periodicity as well. Following this idea, we consider not only the local distance $d_0 = |k^{(t)} - T((j_n + \Delta j)^{(o)})|$, but also two distances separated by one period, that is $d_{-1} = |k^{(t)} - T((j_n + \Delta j - f)^{(o)})|$ and $d_1 = |k^{(t)} - T((j_n + \Delta j + f)^{(o)})|$, where $f$ is the dominant period obtained via Fast Fourier Transform. With the weight for averaging the periodic component being $\omega_{\Delta j} = \frac{1}{\min\{d_{-1},\ d_0,\ d_1\}}$, pattern smoothness can be formulated as

$$\tilde{\boldsymbol{h}}_\tau^{k^{(t)}} = \sum_{\Delta j \in [-f,\ f]} \frac{\omega_{\Delta j}}{\omega} \hat{\boldsymbol{h}}_\tau^{k^{(t)}}\left[(j_n + \Delta j)^{(o)}\right], \quad \text{with} \quad \omega = \sum_{\Delta j \in [-f,\ f]} \omega_{\Delta j}. \tag{5}$$

Rather than directly set $H'$ to be $H$, we call ITF for multiple times based on a schema where the scale factor for each time should no more than 3. For example, in a weekly-to-daily task, we set the ITF schema to $[3L, H]$, which means two cascade ITFs are utilized and mapping $L$ to $3L$ and $3L$ to $H$, respectively. The final outputs with length $H$, denoted as $\boldsymbol{c}_s$ and $\boldsymbol{c}_\tau$, are the high-resolution conditions.

## 3.4 VELOCITY PREDICTOR

The velocity predictor is constructed using a decoder-only Transformer architecture. In comparison to the original design (Vaswani et al., 2017), several modifications have been introduced, including

Rotary Positional Encoding (RoPE) (Su et al., 2024), Pre-Layer Normalization (Pre-LN) (Xiong et al., 2020), and a specially designed cross-resolution attention (CRA) mechanism.

The CRA comprises two sub-layers, *i.e.*, cross-attention to the ITF aligned conditions, and cross-attention to the given low-resolution time series, arranged in a cascade manner. Let $x$ denote the hidden state before CRA, the first CRA layer can be defined as

$$\hat{x} = \begin{cases} \text{CrossAttn}(\text{LayerNorm}(x),\ c_s,\ c_s), & \text{for periodic component;} \\ \text{CrossAttn}(\text{LayerNorm}(x),\ c_\tau,\ c_\tau), & \text{for trend component.} \end{cases} \quad (6)$$

For SSR, the output of the second CRA layer is gated by the target mask sequence $m$, which enables the attention only calculated on the available time steps. The purpose here is to adjust the generated series at the available low-resolution time steps, since the decomposition can deviate $\{d_t | t \in p\}$ from 0. For ASR, we broadcast $l$ to all high-resolution time steps (denoted as $l'$) and use it to calculate the cross-attention with unmasked input. This enables the model to learn an aggregated value for each segment. Based on the above discussion, the second CRA layer can be formulated as

$$y = \begin{cases} m \cdot \text{CrossAttn}(\text{LayerNorm}(\hat{x}),\ l,\ l), & \text{for SSR;} \\ \text{CrossAttn}(\text{LayerNorm}(\hat{x}),\ l',\ l'), & \text{for ASR.} \end{cases} \quad (7)$$

The structure enables the model to first condition on decomposed, high-resolution covariate information, and subsequently modulate the representation using higher-level contextual cues.

## 4 Experiments

To evaluate the usability of SRT, we perform extensive experiment on nine public datasets, which are ETTh1, ETTh2, ETTm1, ETTm2, weather, PEMS-SF, MotorImagery, SelfRegulationSCP1, and SelfRegulationSCP2. Details about data preprocessing are introduced in Appendix. C.

We select eight baselines in relative field, which are SRDiff (Li et al., 2022), ResShift (Yue et al., 2023), IDM (Gao et al., 2023), FlowIE (Zhu et al., 2024), CSDI (Tashiro et al., 2021), FTS-Diffusion (Huang et al., 2024), Diffusion-TS (Yuan & Qiao, 2024), and FlowTS (Hu et al., 2024).[2] Introduction about the baselines and their adaptation to TSSR are summarized in Appendix D.

### 4.1 Results

For comprehensiveness, MSE and DTW distance are employed as evaluation metrics to assess the pointwise error and the overall error, respectively. Table. 1 summarizes the comparative results on three public datasets, while the complete experimental results on all nine datasets are reported in Appendix. H due to space constraints.

Table 1: Quantitative comparison (MSE / DTW distance) results. The best performance ones are bolded and the second best ones are underlined.

| Methods | SSR | | | ASR | | |
|---|---|---|---|---|---|---|
| | ETTm1 | weather | PEMS-SF | ETTm1 | weather | PEMS-SF |
| SRDiff | 0.042 / 0.069 | 0.085 / 0.101 | 0.133 / 0.148 | 0.041 / 0.075 | 0.049 / 0.080 | 0.231 / 0.187 |
| ResShift | 0.040 / 0.071 | 0.081 / 0.089 | 0.186 / 0.161 | 0.044 / 0.078 | 0.047 / 0.075 | 0.224 / 0.195 |
| IDM | 0.036 / 0.064 | 0.039 / 0.045 | 0.108 / 0.072 | 0.037 / **0.068** | 0.092 / 0.095 | 0.126 / 0.079 |
| FlowIE | 0.039 / 0.069 | 0.076 / 0.080 | 0.141 / 0.152 | 0.041 / 0.073 | 0.055 / 0.084 | 0.189 / 0.140 |
| CSDI | 0.037 / 0.063 | 0.034 / **0.028** | 0.109 / 0.072 | 0.040 / 0.072 | 0.224 / 0.073 | 0.128 / 0.074 |
| FTS-Diff | 0.039 / 0.066 | 0.033 / 0.030 | 0.111 / 0.105 | 0.039 / 0.070 | 0.050 / 0.081 | 0.172 / 0.133 |
| Diff-TS | 0.046 / 0.077 | 0.206 / 0.133 | 0.233 / 0.159 | 0.042 / 0.075 | 0.100 / 0.097 | 0.205 / 0.159 |
| FlowTS | 0.036 / 0.067 | 0.106 / 0.093 | 0.122 / 0.117 | 0.041 / 0.074 | 0.176 / 0.131 | 0.385 / 0.230 |
| SRT | **0.026** / **0.057** | **0.031** / 0.039 | **0.097** / **0.070** | **0.037** / 0.069 | **0.035** / **0.068** | **0.125** / **0.073** |

---

[2]Due to space limitations, FTS-Diffusion is abbreviated as FTS-Diff, Diffusion-TS as Diff-TS, SelfRegulationSCP1 as SCP1, and SelfRegulationSCP2 as SCP2. These abbreviations will be used throughout the rest of this paper.

Beyond quantitative evaluations, we further illustrate qualitative results by visualizing the TSSR outputs generated by each model. Figure. 1 presents a comparative analysis between SRT and the two top-performing baseline models, whereas the complete set of results is provided in the Appendix. H.

It can be summarized from Table. 1 that SRT outperform all the baselines *w.r.t.* both MSE and DTW distance. Among the three demonstrated datasets, SRT only lost the top 2 position in SSR task for weather when benchmarked by DTW distance. Moreover, as evidenced by the visualization, SRT generates super-resolution outputs with low levels of noise in the steady-state regions, thereby avoiding excessive volatility. Furthermore, SRT is more capable of reconstructing high-resolution fluctuation details in the peak regions than the baseline models. In summary, SRT, which is specifically tailored for TSSR task, does address this particular problem with great effectiveness.

## 4.2 ABLATION STUDY

In order to validate the effectiveness of each component in SRT, we demonstrate comparable experiments between the full SRT and different model variants. These variations include (1) w/o ITF schema, (2) w/o pattern smoothness in ITF, (3) w/o the whole ITF, (4) w/o RoPE in velocity predictor, (5) w/o Pre-LN in velocity predictor, (6) w/o CRA design in velocity predictor, and (7) w/o the disentanglement based generation. Details about the implementation can be found in Appendix. E.4.Without loss of generality, we implement the ablation study on SSR task, and the performance gaps between the full SRT and different model variants are listed in Table. 2.

Table 2: Ablation study on SSR reporting performance gap with the standard SRT.

| Model Variants | ETTm1 | weather | PEMS-SF |
|---|---|---|---|
| w/o ITF schema | +0.004 / +0.007 | +0.005 / +0.007 | +0.014 / +0.008 |
| w/o pattern smoothness | +0.010 / +0.009 | +0.013 / +0.009 | +0.017 / +0.015 |
| w/o ITF | +0.011 / +0.006 | +0.021 / +0.018 | +0.022 / +0.011 |
| w/o RoPE | +0.008 / +0.005 | +0.005 / +0.007 | +0.032 / +0.019 |
| w/o Pre-LN | +0.009 / +0.008 | +0.018 / +0.011 | +0.003 / +0.005 |
| w/o CRA | | | |
| $\rightarrow$ w/o 1st layer | +0.004 / +0.002 | +0.002 / +0.003 | +0.004 / +0.003 |
| $\rightarrow$ w/o 2nd layer | +0.002 / +0.003 | +0.031 / +0.026 | +0.006 / +0.010 |
| $\rightarrow$ w/o both layers | +0.008 / +0.017 | +0.029 / +0.052 | +0.012 / +0.011 |
| w/o disentanglement | | | |
| $\rightarrow$ *w.r.t.* $d$ | +0.013 / +0.008 | +0.026 / +0.049 | +0.025 / +0.028 |
| $\rightarrow$ *w.r.t.* $l$ | +0.005 / +0.007 | +0.029 / +0.031 | +0.016 / +0.009 |
| $\rightarrow$ *w.r.t.* $d$ and $l$ | +0.016 / +0.013 | +0.048 / +0.047 | +0.046 / +0.073 |

## 4.3 VELOCITY PREDICTOR SELECTION

To verify the efficacy of our design for Velocity Predictor, we replace the proposed velocity predictor by different neural networks with similar level of parameter scale and compare the TSSR performance. These predictor include MLP, TCN, UNet, LSTM, and vanilla Transformer. The results are demonstrated in Table. 3.

Table 3: Velocity predictor selection on SSR reporting performance gap with the standard SRT. '$\Delta$%Param.' denotes the percentage change in the parameter scale of each predictor compared to the proposed velocity predictor in SRT.

| Predictors | ETTm1 | weather | PEMS-SF | $\Delta$%Param. |
|---|---|---|---|---|
| MLP | +0.042 / +0.014 | +0.037 / +0.069 | +0.203 / +0.021 | +14.26% |
| TCN | +0.002 / +0.001 | +0.101 / +0.002 | +0.028 / +0.027 | +0.22% |
| UNet | +0.041 / +0.140 | +0.025 / +0.076 | +0.054 / +0.148 | +10.75% |
| LSTM | +0.013 / +0.019 | +0.131 / +0.055 | +0.031 / +0.013 | +2.01% |
| Transformer | +0.085 / +0.048 | +0.072 / +0.066 | +0.067 / +0.092 | +2.21% |

## 4.4 GENERATION EFFICIENCY

In SRT, we utilize the rectified flow-based generative model to synthesize high-resolution trend and periodic details. A key consideration for this approach is that rectified flow learns a nearly straight transition path from the prior state to the target state, which enables the generation process to be completed with significantly fewer sampling steps. In our model, we apply four-step-sampling, whereas replacing the generative model with DDPM results in inferior outputs even with 200 sampling steps (detailed discussion can be found in Appendix. G.1).

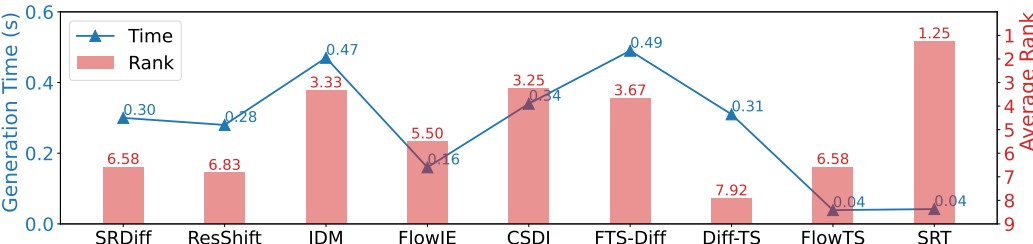

Figure 3: Generation speed and TSSR performance. The curves show the average time required for super-resolution each time-series sample, while the bar chart presents the average performance ranking on the ETTm1, Weather, and PEMS-SF datasets.

Figure. 3 demonstrates the superiority of SRT in generation speed compared to the baseline models. Simultaneously, to evaluate the effectiveness of different models comprehensively, we also report their average performance rankings across three datasets, two tasks, and two metrics in Table. 1. As shown, FlowTS and SRT achieve the top two speeds in generation, substantially outperforming other baseline methods. However, when considering overall TSSR performance, FlowTS attains an average ranking of only 6.58 across the three datasets in Table. 1, which is significantly lower than SRT's ranking of 1.25. These results indicate that SRT not only delivers accurate super-resolution outputs but also benefits from efficient inference speed, which further enhances its practicality. A more comprehensive analysis of model efficiency, including training efficiency and inference FLOPs, can be found in Appendix. E.2.

## 5 EXTENDING TO SRT-LARGE

To enable zero-shot super-resolution capabilities in our model, we modified the standard SRT architecture and conducted extensive pretraining using large-scale datasets from multiple domains, *e.g.*, retail, web search trend, power, and transportation, *etc*, resulting in the SRT-large model. The modifications include increasing the number of attention heads and the hidden dimension of the FFN, and enlarging the number of decoder blocks, bringing the total parameter count to 30 million. Due to considerations regarding the scale of the model and the abundance of training data, we further removed the dropout layers. Additionally, we removed the MLP from the ITF, while retaining temporal enrichment and pattern smoothness. Previously, the condition is directly generated by the ITF. However, with the enhanced generalization capacity of the decoder-base velocity predictor after increasing the parameter count, the model can now directly process coarser condition sequences by inferring the hidden relation inside the networks. Therefore, the value prediction step, which was originally part of the ITF, has been moved into the decoder.

To address the challenge of varying dimensionality across different datasets, SRT-large is implemented as a channel-independent pretrained model for univariate time series, similar approach used in recent models like Lag-Llama (Rasul et al., 2023), TimesFM (Das et al., 2024) and sundial (Liu et al., 2025) have proofed its effectiveness. When performing super-resolution on multivariate time series, SRT-large carries out super-resolution for each dimension separately before combining the outputs for all dimensions.

Different from the intra-dataset evaluation conducted for the standard SRT, cross-dataset evaluation is adopted to better proof SRT-large's zero-shot capability. Figure. 4 summarizes the performance of SRT-large on the SSR task. As shown, SRT-large not only achieves state-of-the-art results across

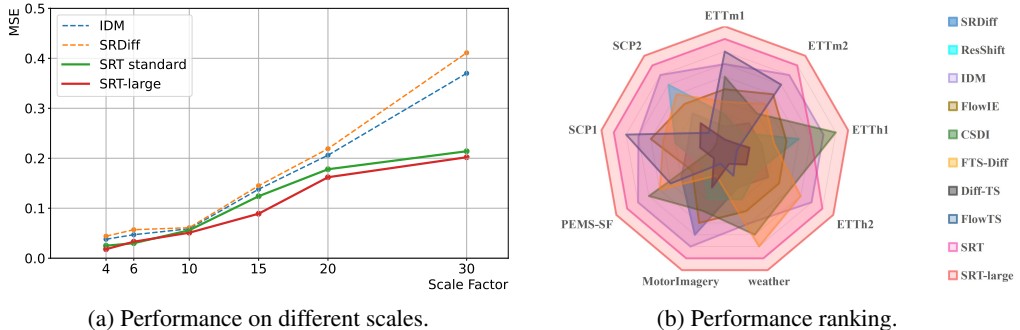

(a) Performance on different scales.  (b) Performance ranking.

Figure 4: Performance of SRT-large on SSR task. (a) Results of SSR tasks on MotorImagery show that SRT-large, similar to SRT, exhibits more stable performance across different scale factors compared to the other two leading baselines. (b) Visualization of method rankings using MSE as the evaluation metric demonstrates that SRT-large consistently achieves top performance across all datasets.

all datasets, but also demonstrates more stable performance than baseline models at different super-resolution scales. A comprehensive presentation of the results for both tasks across the nine datasets is provided in Appendix. H, where qualitative analysis of SRT-large are also available.

It should be noted that the more stable TSSR performance and zero-shot super-resolution capability of SRT-large come at the cost of increased computational demands during inference. This is reflected in higher end-to-end FLOPs, longer average processing time per super-resolved sample, and greater GPU memory requirements (see more in Appendix. E.2). Consequently, the two versions of SRT offer flexible choices for practical applications. In scenarios where historical high-resolution data are available and generation speed is critical, the standard SRT provides highly efficient solution. Conversely, when historical high-resolution data are unavailable and computational resources are sufficient, SRT-large presents a viable solution for achieving highly accurate zero-shot TSSR.

## 6 CONCLUSION

This paper presents SRT, a model specifically designed for time series super-resolution. SRT decomposes the target high-resolution details into periodic and trend components, and employs rectified flow to generate these details separately, conditioned on high-resolution features aligned via Implicit Time Function. To further enhance the velocity modeling within rectified flow, we proposed a cross-resolution attention mechanism, integrated into a decoder-only predictor. In addition, we extend the standard SRT with increased parameters and large-scale pretraining, resulting in SRT-large, which demonstrates strong zero-shot super-resolution capability. Extensive experiments on nine datasets and two subtasks show that both the standard SRT and SRT-large achieve leading super-resolution performance. Moreover, compared with baseline methods, our models provide more stable and high-quality results under various super-resolution scales. These findings highlight the effectiveness and generalizability of SRT in time series modeling, paving the way for future research in this area.

Our approach is built on several implicit structural assumptions, such as the decomposability of both high-frequency and low-frequency sequences, leveraging low-frequency periodicity and trend to guide the generation of corresponding components in the high-frequency residuals. While experiments demonstrate that these priors provide strong modeling advantages, it is important to note that real-world time series data can exhibit a broad range of behaviors, and such assumptions may not always be strictly satisfied. For future work, we aim to further enhance the flexibility and generality of our framework by exploring adaptive methods for prior selection and structural decomposition, allowing the model to better accommodate diverse time series characteristics. We believe that efforts toward more data-driven and universally applicable modeling strategies will expand the practical value of time series super-resolution.

## REPRODUCIBILITY STATEMENT

To facilitate the reproducibility of our work, we have made the following efforts. First, in the main text, we provide detailed descriptions of each module of SRT in the form of narrative explanations, figures, and equations, based on which the model can be reproduced to a large extent. For details that could not be described exhaustively due to space limitations, we provide further clarifications in the appendix. Detailed preprocessing steps to adapt datasets for both SSR and ASR tasks are thoroughly documented in Appendix. C. Descriptions and necessary adaptations of all baseline methods for TSSR task are provided in Appendix. D. Comprehensive implementation details, including hyperparameter settings and key design choices (*e.g.*, interpolation of low-resolution inputs), are specified in Appendix. E.1. The evaluation metrics are elaborated in Appendix. E.3. Implement details about the experimental analysis, including the model modification during ablation study and the network design for each candidate velocity predictor, are comprehensively described in Appendix. E.4 and Appendix. E.5, respectively. We believe that the detailed descriptions provided in the corresponding sections will facilitate readers' reproduction of our work. When we submitted the manuscript for peer review, the complete source code and datasets are provided as supplementary materials.

## ETHICS STATEMENT

This work presents a novel methodological contribution to time series super-resolution. The research is based solely on publicly available benchmark datasets that do not contain personal identifying information. To the best of our knowledge, this study does not raise any ethical issues, as it involves no human subjects, poses no foreseeable risks of misuse, and introduces no apparent biases. The intended applications of this work, such as improving the utility of data in scientific and industrial monitoring, are positively aligned with the goals of ethical research and development.

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

## OUTLINE OF THE APPENDIX

This appendix provides supplementary materials to support the main content of the paper. The structure is organized as follows to facilitate navigation.

In Appendix. A, we disclose our usage of large language models during the writing of this paper. Appendix. B offers a detailed discussion that distinguishes the TSSR task from time series imputation, serving as an extended elaboration on the relevant points raised in the Introduction and Related Work. The data preprocessing procedures for the two TSSR sub-tasks, *i.e.*, SSR and ASR, are then introduced in Appendix. C across two dedicated subsections. Following this, Appendix. D presents the baseline methods used in our experiments and explains the necessary adaptations made for a fair comparison on the TSSR task.

Appendix. E is dedicated to supplementary experimental details. E.1 describes our hyperparameter configuration and E.2 investigates the computational efficiency during training and generation. Two key metrics for evaluating TSSR performance is introduced in E.3. Experimental designs for the ablation study and the velocity predictor selection are further detailed in E.4 and E.5, respectively.

Appendix. F focuses on SRT-large, outlining its specific modifications compared to the standard SRT and describing our pre-training methodology.

We then provide extended discussion in Appendix. G. G.1 discusses the performance advantages of the rectified flow framework in SRT over diffusion-based models. G.2 incorporates additional classic time series interpolation methods for a more comprehensive comparison, presenting both their quantitative and qualitative results against SRT. The contribution of SRT's disentanglement structure to interpretable super-resolution is demonstrated in G.3, followed by a discussion on the out-of-distribution TSSR scenario in G.4.

Finally, the complete set of visualizations and quantitative results, which serve as a full supplement to the condensed presentations in the main text, are compiled in Appendix. H.

# A    LARGE LANGUAGE MODELS USAGE DISCLOSURE

The authors employed large language models (LLMs) solely as a general-purpose tool to assist with the writing process. The model's role was strictly limited to polishing and refining the linguistic expression in parts of the manuscript, including but not limited to improving sentence fluency, adjusting academic tone, and enhancing word choice.

It is crucial to emphasize that the LLMs played no role in the core intellectual contributions of this work. All central ideas, including the research conception, algorithmic design (*e.g.*, the disentangled rectified flow framework, the implicit time function, and the cross-resolution attention mechanism), literature review, theoretical reasoning, implementation of experiments, analysis of results, and drawing of conclusions, were originated and conducted entirely by the human authors. The authors are solely responsible for the accuracy, integrity, and validity of all technical content, claims, and citations presented in this paper.

The LLMs were not used to generate any novel ideas, conduct literature searches, perform data analysis, or create figures and tables. Consistent with ICLR policy, the LLMs are not considered a contributor and are not eligible for authorship. The authors of this paper assume full responsibility for the entire content.

# B    TSSR AGAINST TIME SERIES IMPUTATION

Although both imputation and super-resolution aim to generate missing data points based on known ones, there are still significant differences between the two. These differences can be summarized as the following three main points.

(1) The number of data points need to be generated is different. Typically, imputation generate less points compared to the available data. Experiments setting for most of the work, *e.g.*, TimesNet, set the maximum mask ratio to 50%. On the contrary, TSSR usually generate significantly more data points. When the scale is set to 24, which is a frequently encountered situation when generating hourly series from the daily one, TSSR needs to generate 23 times of the available data volume.

(2) The sampling rate of the imputation task is defined when the time series is given. However, the sampling rate of TSSR varies based on different given scale factors.

(3) The local temporal dependency can be inferred directly from the available data when dealing with the imputation task. In contrast, in super-resolution tasks, since only coarse-grained observations are present and fine-grained data points are missing, it is not feasible to directly deduce local temporal dependencies based solely on the known points. Therefore, it is necessary to mine such dependencies from other known fine-grained segments, as well as from the relationships among different covariates.

# C    DATASETS PREPROCESSING

The nine public datasets we select are ETTm1, ETTm2, ETTh1, ETTh2, weather, PEMS-SF, MotorImagery, SelfRegulationSCP1, and SelfRegulationSCP2. The last three datasets are from the UEA repository, which was originally intended for time series classification. For our experiments, we use data from a single class within each dataset for our super-resolution task.

During the experiments, we use the original series as the high-resolution groundtruth while down sampling the original series to get the low-resolution input. We apply different down sampling policy for SSR and ASR to suit their problem definition.

A train-test split ratio of 8:2 is used. For the three datasets from the UEA repository, we disregard the predefined training and test splits and instead merge the entire dataset before splitting it into new training and test sets according to the aforementioned ratio.

## C.1 Down Sampling for SSR

low-resolution time series are obtained by fixed-step point sampling, which selects the value at a predetermined time step within each interval. This preprocessing keeps aligned with ***Definition 1*** in the main paper. For different datasets, we choose varying sampling time positions within each aggregation window for downsampling, thereby better aligning the processed data with its real-world significance. The detail is summarized in Table. 4.

Table 4: Detail of preprocessing for SSR. The 'Time Position' column indicates the fine-grained time points selected within each coarse-grained interval during the downsampling process.

| Datasets | Scale | Time Position | Dimensions |
|---|---|---|---|
| ETTh1/ETTh2 | 12 | 8:00 and 20:00 everyday | all dimensions |
| ETTm1/ETTm2 | 4 | each whole hour | all dimensions |
| weather | 6 | each whole hour | p, T, rh, VPact, wd, and Tlog |
| PEMS-SF | 6 | each whole hour | [100, 110) |
| MotorImagery | 10 | every 10 data points | {28, 29, 36, 37 } for class 'finger' |
| SelfRegulationSCP1 | 4 | every 4 data points | all dimensions for class 'negativity' |
| SelfRegulationSCP2 | 4 | every 4 data points | all dimensions for class 'negativity' |

## C.2 Down Sampling for ASR

The time series is aggregated by average pooling with both the window size and the stride equal to the scale factor, which suit ***Definition 2*** in the main paper. The detail is summarized in Table. 5. We additionally include the physical meaning corresponding to the super-resolution task in the table. Since the scale factors for both SSR and ASR are consistent within the same dataset, the physical meaning of the SSR task can also be referenced in Table. 5.

Table 5: Detail of preprocessing for ASR. The column 'Interpretation' refers to the physical meaning of the TSSR task base on this preprocessing policy.

| Datasets | Scale | Dimensions | Interpretation |
|---|---|---|---|
| ETTh1/ETTh2 | 12 | all dimensions | twice-daily $\rightarrow$ hourly |
| ETTm1/ETTm2 | 4 | all dimensions | quarterly $\rightarrow$ hourly |
| weather | 6 | p, T, rho, wv, rain, and SWDR | every 10 minutes $\rightarrow$ hourly |
| PEMS-SF | 6 | [100, 110) | every 10 minutes $\rightarrow$ hourly |
| MotorImagery | 10 | {28, 29, 36, 37 } for class 'finger' | 100Hz $\rightarrow$ 1000Hz |
| SelfRegulationSCP1 | 4 | all dimensions for class 'negativity' | 64Hz $\rightarrow$ 256 Hz |
| SelfRegulationSCP2 | 4 | all dimensions for class 'negativity' | 64Hz $\rightarrow$ 256 Hz |

## D  Baselines

### D.1  Image Super-Resolution

**SRDiff.** The method formulates super-resolution as a conditional denoising process, progressively refining noisy inputs into high-resolution residue images under the guidance of the low-resolution image. By leveraging probabilistic modeling and iterative inference, SRDiff generates realistic textures and preserves semantic details, overcoming issues such as over-smoothing and mode collapse commonly observed in prior approaches. The noise predictor is constructed by an UNet with a RRDB-based low-resolution encoder providing the conditional information.

**ResShift.** The method is an efficient diffusion-based approach designed for image super-resolution. Rather than generating the residue images like SRDiff, ResShift constructs the transition between the high-resolution image and the low-resolution image directly. To address the slow inference of traditional diffusion models, ResShift introduces a residual shifting mechanism that effectively

reuses residual information during sampling, significantly reducing the number of steps required and improving generation efficiency while maintaining high reconstruction quality.

**IDM.** The method primarily addresses the problem of continuous super-resolution. IDM integrates implicit neural representation with the diffusion model framework. The core structure of the noise predictor is a UNet, where each upsampling layer incorporates the implicit local image function from LIIF. The continuous coordinates within this function are controlled by the scale factor. By leveraging these continuous coordinates, the network is capable of performing super-resolution across a wide range of scales.

**FlowIE.** The method leverages rectified flow for image enhancement where image super-resolution is an important component. The rectified flow enables fast sampling by abandoning the extensive denoising steps compared with the aforementioned diffusion-based models. Besides, FlowIE proposes a novel many-to-one transport mapping, which bridges between any noise to one real-world image. This design addresses the limitation that the one-to-one mapping in the original rectified flow may be influenced a lot by the gap between the synthetic and the real-world images.

**For all the above methods,** we aligned the dimensions of the time series data to those of the image data in our experiments. Specifically, the batch dimension (B) remains unchanged; the temporal dimension (T) of the time series is mapped to the width (W) of the image, and the feature dimension (D) is mapped to the height (H) of the image. The channel dimension (C) is set to 1.

## D.2 Time Series Generative Models

**CSDI.** The method is designed to address the challenge of missing value imputation in multivariate time series data. By utilizing conditional score-based diffusion models, CSDI explicitly models the conditional distribution of missing data given observed values and arbitrary missing patterns. The method applies a denoising diffusion process conditioned on available observations and flexible masking, enabling probabilistic and diverse imputations. Since cSDI is capable of generating missing points controlled by masking, in our experiments, we modify the mask to let the model generate the unknown high-resolution points between consecutive available points.

**FTS-Diffusion.** The method is designed specifically for financial time series generation. It is composed of three modules, *i.e.*, pattern recognition, pattern generation, and pattern evolution. The pattern recognition module extract the recursively shown scale-invariant patterns which are latter fed into diffusion models to train the pattern generation module. Finally, the pattern evolution module determines the temporal transition of the generated patterns, so the whole series can be recursively concatenated one subseries after another. In our experiments, we keep the pattern recognition and pattern generation as the original FTS-Diffusion and modify the pattern evolution module to suit the TSSR task. The original version predict next generated subseries based on the last known segment using a three-layer-MLP. We change the prediction input from the last known segment to the given low-resolution time series and keep the architecture of the prediction network unchanged.

**Diffusion-TS.** The model utilizes diffusion model for time series generation. Similar to our proposed SRT, Diffusion-TS also enroll time series decomposition. However, the decomposition of Diffusion-TS is implemented inside the transformer-based model while we choose to use two disentangled rectified flow models for parallel generation of the two components. Besides, Diffusion-TS can handle conditional generation based on the unconditional model by incorporating gradient guidance. In our experiment, we leverage the conditional capability of Diffusion-TS by using the steps from the low-resolution time series as conditional parts and the high-resolution steps to be generated as the generative parts.

**FlowTS.** The method shares the high-level generative model similar to our proposed method by leveraging rectified flow to realize efficiency time series generation. FlowTS can handle both unconditional and conditional generation via adaptive sampling. In our experiments, the original training part is remained but the sampling policy for conditional generation is modified. The proposed adaptive sampling, summarized as Algorithm 2 in the FlowTS paper, refines the predicted state by the partially observed target. For the adaptation to TSSR task, we set the low-resolution series as the partially observed target and use our target indexing sequence $M$ to initialize the observation mask.

# E  EXPERIMENTAL DETAILS

## E.1  HYPERPARAMETER SETTINGS AND SENSITIVITY ANALYSIS

There are two instances of time series decomposition in our model, *i.e.*, the decomposition of $d$ into $s$ and $\tau$, and the decomposition of $l$ into $l_s$ and $l_t$. In our parameter settings, both decompositions employ average pooling with the same kernel size, *i.e.*, $kernel_d = kernel_l$, which is dataset-dependent. For ETTh1 and ETTh2, the kernel size is set to 7. For MotorImagery, the kernel size is 13, and for all other datasets, the kernel size is set to 25.

For generating $l^*$ from $l$ through interpolation, we used linear interpolation. If $l$ is a multivariate time series, linear interpolation is performed independently for each dimension to upsample the low-resolution data to the length of the high-resolution sequence.

During the temporal enrichment stage of ITF, the hyperparameter $r$ is set to 3. For the value prediction stage, we employ a two-layer MLP as the predictor, with a hidden layer dimension of 128.

Our proposed velocity predictor is constructed in a decoder-only manner, consisting of 3 decoder blocks. Each decoder block employs four heads for both self-attention and CRA and a feed forward network with the dimension of 128. The input dimension to each decoder is 128, which is obtained by the input embedding and condition embedding. For the entire predictor, the input dimension corresponds to the dimensionality of the time series to be processed, while the condition dimension is set to twice the time series dimensionality.

When training, the batch size is set to be 32 and the initial learning rate to be 1E-3 with 80% decay every 10 steps. During testing, we use a 8:2 training-testing split ratio for all the ten public datasets.

While the aforementioned hyperparameter values are determined through empirical optimization, sensitivity analysis *w.r.t.* five key hyperparameters is further conducted to assess the robustness of our selections. The analysis is performed by systematically varying each parameter within a reasonable range around the recommended values, while keeping others fixed at their optimal settings.

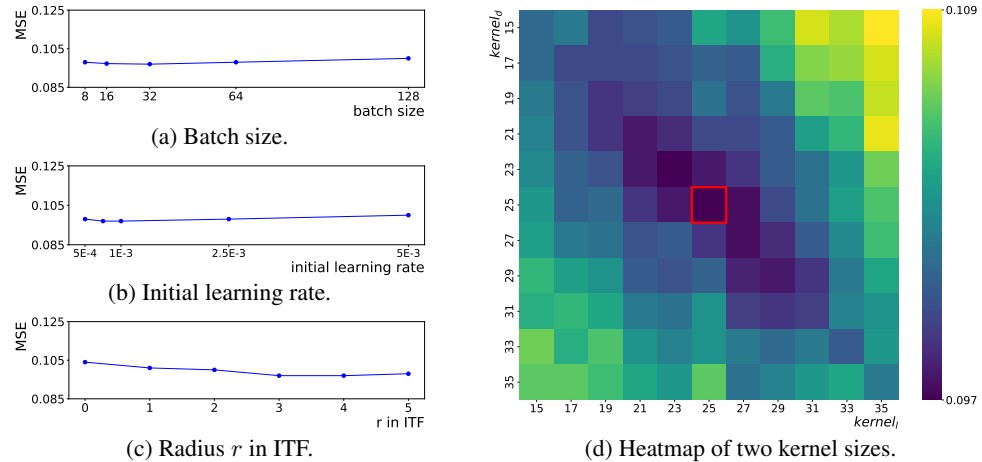

Figure 5: Sensitivity analysis of five key hyperparamteters on PEMS-SF dataset.

As illustrated in Figure. 5, the sensitivity curves for individual parameters (Figure.5a - 5c) demonstrate remarkable stability across substantial parameter variations. The heatmap analysis of kernel size combinations (Figure.5d) reveals an interesting symmetry property, *i.e.*, optimal performance occurs when $kernel_d = kernel_l$. Besides, even with the most suboptimal hyperparameter combination, SRT still achieves the second-best performance among all baselines, trailing IDM by only 0.001 in terms of MSE.

The observed robustness across multiple hyperparameters significantly enhances the practical utility of our approach, as it reduces the burden of precise parameter tuning and ensures consistent performance across different implementations.

### E.2 Training and Generation Efficiency

**Training.** Some of the rectified flow variations leverage the reflow operation to make the learned state transition path more linear, enabling fast inference speed with fewer steps or even single-step generation. In SRT, we balance the efficiency between training and generation. We omit the reflow operation, and instead employ four-step generation in the inference phase rather than single-step generation.

**Generation.** Under SRT's disentangled rectified flow architecture, a decoder-only predictor is used to produce the velocity at each step, based on which the state transitions are executed in four steps. In this process, two randomly initialized noise vectors are respectively mapped to high-resolution periodic and trend details. Although the theoretical foundations are complex, the value prediction network adopted by ITF is simple. Moreover, the decoder-only structure based velocity predictor is involved in only four forward passes. As a result, SRT demonstrates substantial efficiency advantages compared to most baselines. Furthermore, since SRT is inherently designed for the TSSR task, its output requires no further postprocessing, which further amplifies its advantage *w.r.t.* generation efficiency.

With the NVIDIA A100 GPU and CUDA 12.2, we evaluate the efficiency of SRT and other baselines on the ETTm1, Weather, and PEMS-SF datasets. The average training and inference time is summarized in Table. 6.

Table 6: Training and generation time comparison. Time for training time is measured per epoch, while time for generation is measured per sample, and both in seconds.

|  | SRDiff | ResShift | IDM | FlowIE | CSDI | FTS-Diff | Diff-TS | FlowTS | SRT | SRT-large |
|---|---|---|---|---|---|---|---|---|---|---|
| Training Time | 2.70 | 3.03 | 3.84 | 3.35 | 1.75 | 2.97 | 1.82 | 1.53 | 1.77 | - |
| Generation Time | 0.30 | 0.28 | 0.47 | 0.16 | 0.34 | 0.49 | 0.31 | 0.04 | 0.04 | 0.13 |

Additionally, to obtain a standardized and device-independent measure of model complexity, we performed a Floating Point Operations (FLOPs) analysis for the inference stage. Unlike generation time, which is affected by system overhead, FLOPs solely quantifies the computational operations. Using a super-resolution input time series of size [1, 337, 7], we summarize the total FLOPs for each baseline over the entire generation process, including multi-step sampling as well as any preprocessing and postprocessing procedures used, in Table. 7.

Table 7: FLOPs comparison over the entire generation process. The GFLOPs values reported in the table indicate the number of floating point operations in billions, *i.e.*, 1 GFLOP = $10^9$ FLOPs.

|  | SRDiff | ResShift | IDM | FlowIE | CSDI | FTS-Diff | Diff-TS | FlowTS | SRT | SRT-large |
|---|---|---|---|---|---|---|---|---|---|---|
| GFLOPs | 251.01 | 204.17 | 317.16 | 14.39 | 230.63 | 288.03 | 187.49 | 3.74 | 2.49 | 35.46 |

We observe that diffusion-based methods, whether for time series or image super-resolution tasks, exhibit significantly slower inference times and much larger GFLOPs compared to SRT. Although FlowIE is designed based on rectified flow, it is an image super-resolution model that performs upscaling along both the width and height dimensions, and requires additional postprocessing to ensure that the resolution along the channel dimension remains unchanged after adapting the input to time series. As a result, its generation speed is still slower than SRT's. FlowTS, benefiting from adaptive sampling and unconditional training and not requiring extra postprocessing, achieves the fastest training and generation speed among all compared methods. However, its time series super-resolution performance is inferior to that of SRT.

As for SRT-large, although its parameter scale is substantially increased compared to the standard version, it remains relatively modest when contrasted with language models or multi-modal models. Furthermore, owing to its specialized design for the TSSR task, SRT-large contains fewer parameters than general-purpose pretrained models for time series forecasting. In training SRT-large, we did not employ much training techniques, but instead simply applied mixed-precision training and utilized gradient accumulation to avoid excessively small batch sizes. As a result, it costs around 741.21 seconds of average training time per epoch for SRT-large. In the generation stage, due to the channel-independent design, SRT-large has to upscale each dimension separately, which also

drags the generation efficiency compared with the standard version. Besides generation time and FLOPs analysis, peak GPU requirements are also investigated. When using the same [1, 337, 7]-sized pseudo input, the peak GPU memory consumption for SRT is 0.03 GB, compared with SRT-large's 0.26 GB. This substantial increase may drag the total generation speed in practical scenarios. Though the inference can be conducted using non-top-tier GPUs like V100, a smaller batch size has to be settled. For example, on big datasets like MotorImagery, the maximum batch size allowed is 8, which directly impacts the total TSSR processing time.

In summary, the two variants of SRT are suited to different practical scenarios. The standard version is preferable when historical high-resolution data are available and rapid generation is required, while the large version is more appropriate for zero-shot TSSR tasks in situations where computational resources are sufficient.

### E.3 METRICS

In our experiments, we employ both the mean squared error (MSE) and the dynamic time warping (DTW) distance as evaluation metrics. MSE captures the point-wise differences between the generated and target sequences, while DTW measures the overall similarity allowing for temporal misalignment. For both metrics, we treat each batch and channel independently — that is, for each sequence along the time dimension, we compute the metric separately, and then report the average value over all batch and channel combinations.

**MSE** is a commonly used metric for measuring the average point-wise difference between predicted and target sequences. It is computed by averaging the squared differences between the two sequences across all time steps. Formally, given the predicted sequence $\hat{y} = [\hat{y}_1, \hat{y}_2, ..., \hat{y}_n]$ and the ground truth sequence $y = [y_1, y_2, ..., y_n]$, MSE is defined as

$$MSE(x, \ y) = \frac{1}{n} \sum_{i=1}^{n} (\hat{y}_i, \ y_i)^2.$$

MSE provides a straightforward measure of the generation accuracy at each time point, which is suitable to be used as one of the TSSR metrics.

**DTW distance** is a distance measure for time series that allows flexible alignment between sequences, making it robust to temporal phase differences. DTW works by building a cost matrix using pairwise distances and finding the minimum cumulative path through this matrix. The DTW distance between sequences $x = [x_1, x_2, ..., x_n]$ and $y = [y_1, y_2, ..., y_m]$ is formally defined as

$$DTW(x, \ y) = \min_p \sum_{(i,j) \in p} d(x_i, \ y_j),$$

where $p$ denotes a warping path and $d(\cdot, \cdot)$ is the squared Euclidean distance.

The main advantage of DTW distance is its robustness to temporal distortions between two sequences, making it particularly suitable for measuring the similarity between the groundtruth and the super-resolution result. To mitigate the effect of sequence length on the DTW metric, we use the DTW distance normalized by the length of the warping path.

### E.4 ABLATION STUDY

**w/o ITF schema** means we only call ITF for one time rather than aligning the series for multiple times in a recursive manner. We conduct this model variation by setting the ITF schema to $[H]$.

**w/o pattern smoothness** means the prediction from Eq. 3 is used directly as the final ITF result, with the smoothness for both trend and periodic components performed in Eq. 4 and Eq. 5 being skipped.

**w/o ITF** means we deprecate the whole ITF and use the low-resolution series $l$ directly as the condition for the rectified flow process. For temporal alignment, we use the linear interpolation of $l$ to replace the $c_s$ and $c_\tau$ in Eq. 6.

**w/o RoPE** means we give up RoPE for positional encoding in the proposed velocity predictor. Instead, the sinusoidal positional encoding in the vanilla Transformer model is utilized.

**w/o Pre-LN** simply means the Pre-LN in the proposed Velocity Predictor is replaced by the Post-LN used in the vanilla Transformer model.

**w/o CRA** consists of three variants, *i.e.*, without the first layer (case 2), without the second layer (case 2), and without both layers (case 3). For case 1, the cross attention with the ITF output is skipped and Eq. 6 is replaced by $\hat{x} = x$. For case 2, the cross attention with the low-resolution input is skipped and Eq. 7 of the proposed model is replaced by $y = \hat{x}$ For case 3, both of the cross attention layers are no need to be performed and the whole CRA mechanism is replaced by $y = x$.

**w/o disentanglement** consists of three variants, *i.e.*, without the disentanglement of $d$ (case 1), without the disentanglement of $l$ in ITF (case 2), and without both of them (case 3). For case 1, the generation target of the rectified flow changes from $s$ and $\tau$ to $d$, which means the proposed two disentangled rectified flow structure is replaced by one. Formally, the training of the Velocity Predictor is replaced by

$$\min \int_0^1 \mathbb{E}[(d_1 - d_0 - V_d(d_t, t, c, l; \theta_d))^2 dt,$$

where $c = c_s + c_\tau$. The generation process is replaced by

$$\hat{d}_{k_{i+1}} = \hat{d}_{k_i} + (k_{i+1} - k_i)V_d(\hat{d}_{k_i}, k_i, c, l; \theta_d).$$

The final result is generated via

$$\hat{h} = l^* + \hat{d}, \quad \text{with } \hat{d} = \hat{d}_4.$$

For case 2, $l$ no longer needs to be decomposed into $l_s$ and $l_\tau$. Instead, the ITF is applied directly on $l$, aligning the length from $L$ to $H$, whose output is denoted as $c_l$. The first layer of CRA is modified to suit this model variant, as

$$\hat{x} = \text{CrossAttn}(\text{LayerNorm}(x), \ c_l, \ c_l).$$

For case 3, both of the aforementioned modifications *w.r.t.* case 1 and case 2 are implemented.

### E.5 VELOCITY PREDICTOR SELECTION

For fairness, all the candidate predictor have roughly the same level of parameter scale. We list the structure of these models as follows.

**MLP:** The network is composed of 5 MLP layers, each with 512 hidden units.

**TCN:** The network consists of 4 temporal convolutional blocks, with hidden channel sizes of 64, 128, 256, and 512, respectively. Each convolutional block is composed of two dilated convolutions. For the $i$-th block, the dilation is set to be $2^i$.

**UNet:** We adopt a UNet architecture for time series modeling, where the input sequence, condition, and time step are concatenated along the channel dimension and jointly processed. The encoder consists of stacked 1D convolutional blocks with max-pooling for hierarchical feature extraction and progressive temporal downsampling, followed by a bottleneck convolutional block. The decoder restores temporal resolution via transposed convolutions, using skip connections to combine encoder features at each stage. The downsampling channel sizes are 32 and 128, with reversed sizes during upsampling, and the bottleneck channel size is 1024.

**LSTM:** The network comprises four LSTM blocks with hidden dimensions of 64, 128, 256, and 128, respectively. The final output is produced by a fully connected layer that maps the output of the last LSTM block to the same dimensionality as the input.

**Transformer:** We employ a vanilla Transformer architecture. At each time step t, the condition aligned by the ITF is concatenated with the encoder input along the channel dimension before being fed into the model. During training, teacher forcing is utilized, whereas during inference, the decoder's previously generated outputs are recursively used as the inputs for subsequent decoding steps. The model architecture features a hidden layer dimension of 64, 4 attention heads, and 6 layers in both the encoder and decoder.

# F  SRT-LARGE

## F.1  ARCHITECTURE

As discussed in the main text, we increased the model's parameter count to enable effective zero-shot super-resolution of time series, leveraging the increased capacity of larger neural architectures to facilitate the learning of more general and transferable representations during pretraining.

Compared to the standard SRT, we significantly increased the model's parameter count in SRT-large by scaling several architectural components: the number of attention heads in the velocity predictor is increased from 4 to 16; the hidden dimension of the feed-forward networks is expanded from 128 to 512; the input embedding dimension is raised from 128 to 512; and the number of decoder blocks is increased from 3 to 8. Moreover, we discarded the use of dropout in the velocity predictor.

Furthermore, we eliminate the separate predictor in ITF and consolidate its prediction functionality into the velocity predictor. In the original ITF workflow, before pattern smoothness, predictions are made separately for the high-resolution time points corresponding to different low-resolution time points within an examined interval (which, for periodic sequences, encompasses one Fourier period on each side, and for non-periodic sequences, includes one time point on either side). These predictions are based on temporally enriched low-resolution points as well as the time differences between high- and low-resolution sequences to estimate fine-grained values. In SRT-large, we merge the ITF predictor with the velocity predictor, allowing us to pretrain only a single model. With this new architecture, pattern smoothness in ITF can be regarded as directly weighted smoothing over the enriched low-resolution time series.

Other components of the standard SRT, such as time series decomposition and the rectified flow generation, as well as the overall workflow, are retained.

## F.2  PRETRAINING DETAILS

To ensure the diversity of pretraining data, we curate a collection of time series exhibiting various characteristics, including different types of periodicity, trend, and step patterns. These characteristics cover both high-resolution and low-resolution variants, such as high-frequency and low-frequency periodic components. To achieve comprehensive coverage, pretraining data is collected from multiple domains. Additionally, recognizing that real-world data may lack certain temporal patterns, we further augment the pretraining dataset with synthetic time series specifically generated to enrich its representational capacity and to ensure the presence of a broad range of patterns.

**Equity.** We select two three-day trading periods, namely June 10–12, 2025 and June 30–July 2, 2025. Stocks with no trading activity during these periods being excluded. Subsequently, we randomly sample 100 stocks from the S&P 500 index and 400 stocks from the Russell 2000 index. For the selected stocks, we collect intraday price data at the one-minute interval during regular trading sessions across these three days.

**Commodity.** We select daily prices of the main futures contracts for gold, silver, copper, crude oil, natural gas, corn, wheat, and soybean. Additionally, we incorporate daily spot prices for gold, crude oil, and natural gas. For all commodities, data is collected for all trading days within the period from June 15, 2022 to June 13, 2025.

**Currency rate.** We select daily exchange rates of major currency pairs against the US dollar, including EUR/USD, JPY/USD, GBP/USD, CNY/USD, AUD/USD, CAD/USD, and CHF/USD. Historical data from June 15, 2020 to June 13, 2025 is collected, corresponding to trading days within this period.

**Power.** The public dataset 'ElectricityLoadDiagrams20112014' [3] is utilized. We follow the idea of Informer and preprocess the dataset into hourly format.

**Retail.** The public dataset 'Online Retail II' [4] is utilized. We divide the time series into two groups based on country: those with Country equal to United Kingdom and those designated as Others. For

---

[3]https://archive.ics.uci.edu/dataset/321/electricityloaddiagrams20112014
[4]https://archive.ics.uci.edu/dataset/502/online+retail+ii

each group, we aggregated the total hourly sales of all products, retaining only the hours from 9:00 to 17:00 each day.

**Transportation.** The public dataset 'traffic' [5] as well as PEMS-SF (with dimension [100, 110] excluded) are utilized.

**Web search trends.** We utilize Google Trends to identify the top 1,000 most-searched keywords over the past six months. For each query, we collect daily search interest data spanning from December 1, 2015 to June 20, 2025. From these, we select the top 200 ones exhibiting the lowest temporal sparsity for inclusion in our pretraining dataset.

**Synthetic time series.** Finally, to ensure the presence of more typical temporal dependency patterns in the pretraining data, we additionally incorporate artificially synthesized time series. The synthetic sequences mainly consist of three components: (1) a sequence-level trend, generated either by sine functions with random periods (greater than the sequence length) and random phase shifts, or by linear functions; (2) ARIMA sequences parameterized by randomly chosen p, d, and q; and (3) the sum of multiple sine functions with randomly selected frequencies and phase shifts. For each synthetic series, we select random combinations of these three components to generate the final sequence, with the sequence length fixed at 5000.

# G  DISCUSSION

## G.1  RECTIFIED FLOW AGAINST DIFFUSION MODELS

Traditional diffusion models generate samples by simulating a stochastic differential equation (SDE), which gradually transforms Gaussian noise into complex data through a sequence of random perturbations and denoising steps

$$dx = f(x, t)dt + g(t)dW_t,$$

where $dW_t$ denotes a Wiener process.

In contrast, rectified flow models formulate the data transformation as an ordinary differential equation (ODE) governed by a learned vector field, as

$$dx = v(x, t)dt.$$

This ODE-based trajectory provides a deterministic mapping between the data and noise distributions, enabling more efficient and stable sampling without the requirement for stochasticity during the generation process.

Table 8: Quantitative comparison between rectified flow based model and DDPM based model.

| Generators | Steps | SSR | | | ASR | | |
|---|---|---|---|---|---|---|---|
| | | ETTm2 | weather | PEMS-SF | ETTm1 | weather | PEMS-SF |
| rectified flow | 4 | 0.026 / 0.057 | 0.031 / 0.039 | 0.097 / 0.070 | 0.037 / 0.069 | 0.035 / 0.058 | 0.125 / 0.073 |
| DDPM | 4 | 1.307 / 1.228 | 1.896 / 1.704 | 1.975 / 2.020 | 1.968 / 1.225 | 1.281 / 1.593 | 2.396 / 1.959 |
| DDPM | 50 | 0.625 / 0.330 | 0.449 / 0.437 | 0.608 / 0.519 | 0.437 / 0.502 | 0.498 / 0.305 | 0.767 / 0.692 |
| DDPM | 100 | 0.104 / 0.093 | 0.097 / 0.086 | 0.238 / 0.125 | 0.113 / 0.128 | 0.133 / 0.092 | 0.325 / 0.197 |
| DDPM | 200 | 0.031 / 0.062 | 0.036 / 0.044 | 0.119 / 0.072 | 0.036 / 0.075 | 0.047 / 0.069 | 0.173 / 0.102 |

One of the main advantages of rectified flow compared to traditional diffusion models is its ability to generate samples of comparable or even higher quality with significantly fewer sampling steps, thereby improving both generation speed and efficiency. For validation, we replace the rectified flow based generation of $s$ and $\tau$ in SRT with a DDPM based approach, and conduct experiments on the two sub-tasks on ETTm1, Weather, and PEMS-SF. As shown by the quantitative results in Table. 8, when generating samples with only four steps, the performance gap between the DDPM based models and the rectified flow based models is substantial. It is only when the number of DDPM sampling steps increases to 200 that the performance approaches that of the rectified flow based model.

---

[5]https://pems.dot.ca.gov/

## G.2    COMPARISON WITH CLASSIC TIME SERIES INTERPOLATION METHODS

We also include several classic time series interpolation methods for comparison, *i.e.*, nearest neighbor interpolation, linear interpolation, and cubic spline interpolation. Although these methods are relatively basic, their advantages lie in fast computation and deterministic input-output mapping. We adopt the same shuffling scheme as used in Table. 1, and conduct experiments on the ETTm1, Weather, and PEMS-SF datasets for both SSR and ASR tasks. The quantitative results are presented in Table. 9 [6].

Table 9: Quantitative comparison between SRT and three classic time series interpolation methods.

| Methods | SSR | | | ASR | | |
|---|---|---|---|---|---|---|
| | ETTm1 | weather | PEMS-SF | ETTm1 | weather | PEMS-SF |
| NN | 0.070 / 0.059 | 0.066 / 0.056 | 0.138 / 0.093 | 0.053 / 0.084 | 0.059 / 0.080 | 0.150 / 0.119 |
| Linear | 0.046 / 0.065 | 0.051 / 0.041 | 0.099 / 0.087 | 0.053 / 0.079 | 0.063 / 0.085 | 0.146 / 0.096 |
| Spline | 0.051 / 0.064 | 0.059 / **0.037** | 0.101 / 0.086 | 0.052 / 0.075 | 0.062 / 0.091 | 0.146 / 0.093 |
| SRT | **0.026 / 0.057** | **0.031 / 0.039** | **0.097 / 0.070** | **0.037 / 0.069** | **0.035 / 0.068** | **0.125 / 0.073** |

In addition to the quantitative results, we also conducted a visual comparison of the super-resolution outputs from these three methods with those from SRT. The results are shown in Figure. 6.

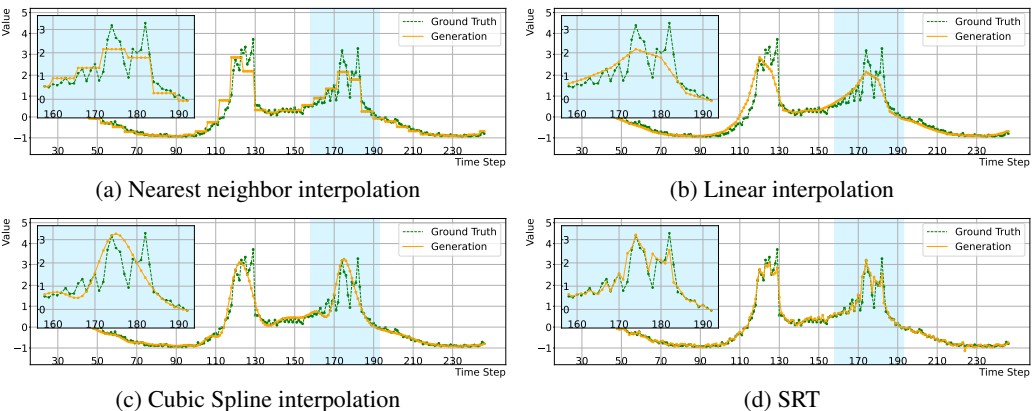

(a) Nearest neighbor interpolation                    (b) Linear interpolation

(c) Cubic Spline interpolation                    (d) SRT

Figure 6: Visualization comparing three classic interpolation methods and SRT.

It can be observed that SRT demonstrates a clear advantage in both SSR and ASR tasks. The visualization of super-resolution results reveals that both nearest neighbor interpolation and linear interpolation can locally deviate significantly from the high-resolution ground truth, leading to distorted outputs. Although cubic spline interpolation is able to better capture the overall trend of time series, its intrinsic mechanism of performing interpolation via polynomial fitting of the known low-resolution data points results in over-smoothed outputs and loss of sharp details around the peak.

## G.3    INTERPRETABILITY ANALYSIS

One of the key advantages of SRT's disentangled architecture is its inherent interpretability. Unlike end-to-end black-box models, SRT explicitly decomposes the super-resolution task into two semantically distinct sub-tasks, *i.e.*, generating the high-resolution detailed periodicity and trend. Rather than focusing solely on the TSSR result, this design helps answer the question of what contributes to the final output.

To visually demonstrate the link between the generated components and the final output, we present a case study on the segment where we demonstrate the qualitative results, as in Figure. 7. Specifically, we visualize the generated periodic and trend detail, and show how their summation with the

---

[6]Nearest neighbor interpolation, linear interpolation, and cubic spline interpolation are abbreviated as NN, Linear, and Spline, respectively.

interpolated low-resolution series. This process reveals the generated trend detail contributes to the long-term shifts, while the generated periodicity contributes to the short-term fluctuations.

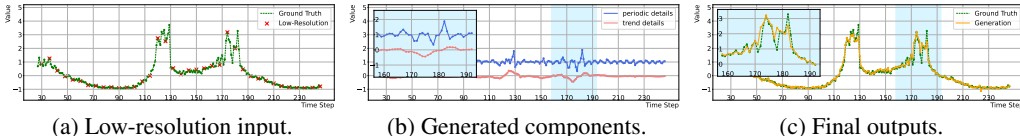

(a) Low-resolution input.      (b) Generated components.      (c) Final outputs.

Figure 7: Visualization of the disentangled TSSR process. Note that the generated periodic component is considered to have an upward offset added to it, in order to avoid overlapping.

The disentangled generation process of SRT not only enhances transparency but also provides a practical mechanism for confidence assessment in real-world applications. By inspecting the independently generated trend and periodic components, a domain expert can leverage their prior knowledge to evaluate the confidence level of each constituent part, thereby forming a more informed judgment about the overall reliability of the super-resolved output.

For instance, consider the application in electrocardiogram super-resolution. A cardiologist examining the results would expect the periodic component to exhibit physiologically plausible P-wave, QRS-complex, and T-wave morphologies. If the generated periodic details deviated significantly from these expected patterns *e.g.*, exhibiting spurious, high-frequency noise or impossible shapes, it would immediately cast doubt on the result's validity. On the other hand, a trend component that shows a gradual heart rate acceleration in a stress test scenario would be deemed more credible than one with erratic, non-physiological jumps. This judgment of the confidence level is a direct benefit of our disentangled architecture, offering better practicality when ground truth is unavailable.

### G.4 SPECIAL CASES

One non-standard application scenario is out-of-distribution super-resolution, *e.g.*, generating weekly data based on the monthly data. Such problems fall within the scope of temporal alignment, rather than being addressed by standard time series super-resolution techniques. As a result, our definitions for SSR and ASR do not encompass these situations.

Nonetheless, SRT can effectively tackle such problems. Taking the example of the aforementioned problem, SRT can first generate daily data from the monthly, and then aggregate the daily data into weekly values. Despite the varying number of days in each month, SRT utilizes a target mask to specify the sampling points at the end of each month, enabling the super-resolution process to be completed. It should be noted that, in contrast to standard TSSR tasks, the difference between consecutive values in the mask index sequence may exhibit multiple distinct values in this scenario.

We design and conduct a supplementary experiment to validate the above capability. We first generate three daily time series with different underlying characteristics using the synthetic data construction method described in Appendix. F.2. These synthesized series, referred to as Toy #1, Toy #2, and Toy #3, constitute our toy dataset for controlled experiments. Specifically, we construct time series that are characterized by long-period periodicity (Toy #1), short-period periodicity (Toy #2), and strong autoregressive behavior (Toy #3), respectively. Each series is assigned virtual dates ranging from January 3, 2000 to June 1, 2025. Next, the daily data are aggregated into weekly and monthly frequency, using both sampling and averaging approaches.

Table 10: Quantitative comparison for out-of-distribution TSSR.

| Methods | SSR | | | ASR | | |
|---|---|---|---|---|---|---|
| | Toy #1 | Toy #2 | Toy #3 | Toy #1 | Toy #2 | Toy #3 |
| IDM | 0.014 / 0.037 | 0.018 / 0.033 | 0.021 / 0.024 | 0.019 / 0.041 | 0.019 / 0.035 | 0.035 / 0.034 |
| FTS-Diff | 0.019 / 0.036 | 0.020 / 0.031 | 0.020 / 0.025 | 0.017 / 0.036 | 0.027 / 0.044 | 0.031 / 0.036 |
| SRT | 0.011 / 0.027 | 0.020 / 0.027 | **0.015** / 0.022 | 0.014 / 0.024 | **0.016** / 0.030 | 0.027 / **0.024** |
| SRT-large | **0.008** / **0.016** | **0.016** / **0.024** | 0.019 / **0.022** | **0.013** / **0.021** | 0.017 / **0.027** | **0.024** / 0.028 |

We then compared the standard SRT and SRT-large models with IDM and FTS-Diff. IDM is designed to emphasize out-of-distribution super-resolution capabilities, while FTS-Diff is one of the state-of-the-art models for time series generation. From the quantitative comparison demonstrated in Table. 10, we can summarize that SRT and SRT-large do tackle the out-of-distribution TSSR tasks more effective than the other two leading baselines.

## H    SUPPLEMENTARY EXPERIMENTAL RESULTS

Due to space limitations in the main text, only a subset of the results was presented. This section provides all the supplementary experimental results to demonstrate the comprehensiveness of the experiments.

The qualitative SSR results for all eight baselines, as well as the two versions of our proposed SRT, are visualized in Figure. 8. Of all the TSSR results, SRDiff, ResShift, FlowIE, and CSDI suffer great oversmoothness during the peak phase. FTS-Diff and IDM produce more detailed pattern than the aforementioned methods, but they fail to reconstruct the double-peak structure that lost by the downsampling. Diff-TS and FlowTS are capable of perceiving volatility changes during the peak phase but perform worse during the stable phase due to their tendency to generate overly volatile sequences. In comparison, both SRT and SRT-large demonstrate the overall best TSSR performance across both stable and peak phases.

The quantitative comparable results on all the nine public datasets are demonstrated in Table. 11 and Table. 12 for SSR and ASR, respectively. Given the large size of the following tables, they are presented in landscape orientation to ensure all columns and information are clearly displayed.

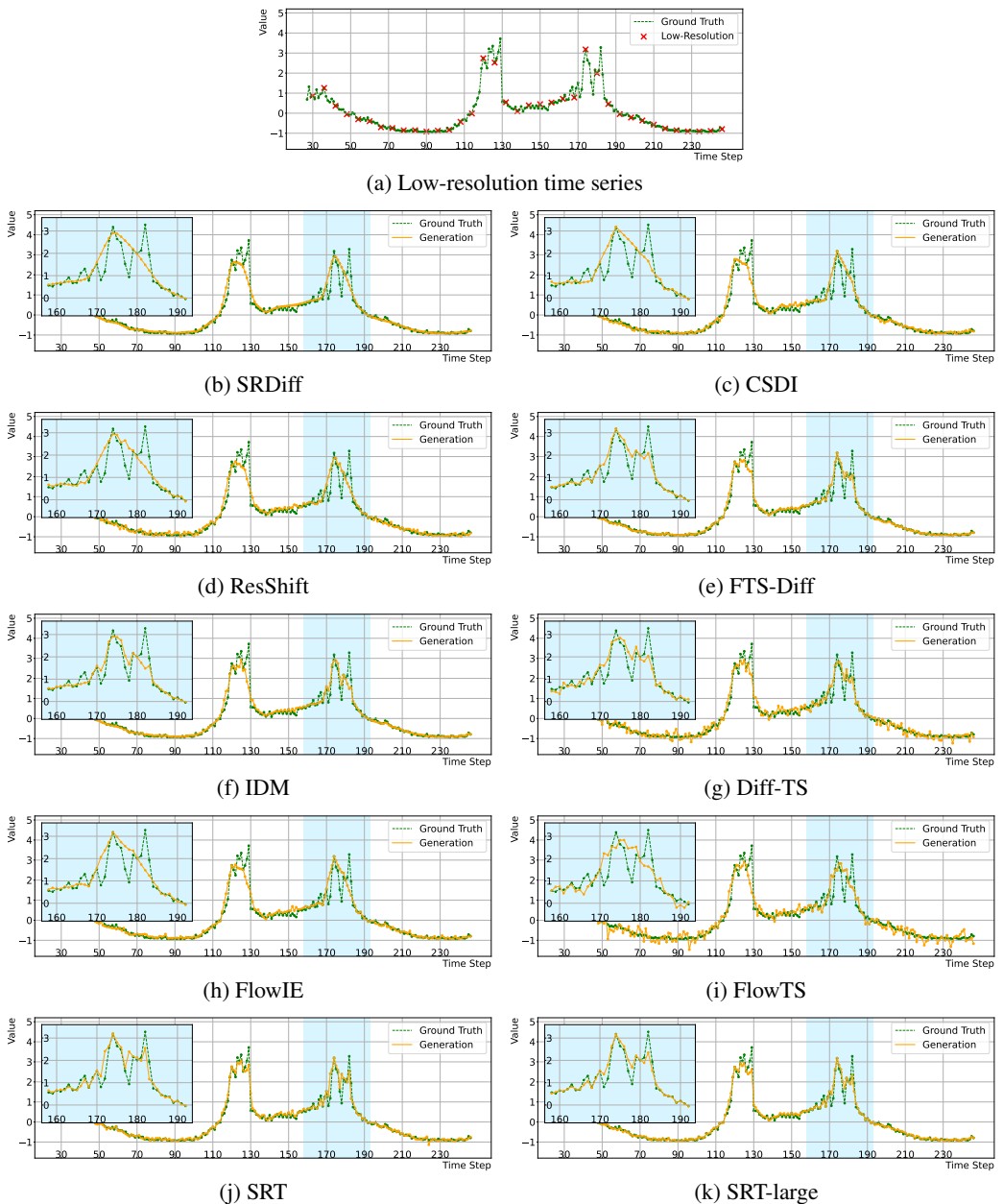

Figure 8: Qualitative analysis on a segment from the PEMS-SF dataset. SRT and SRT-large outperform other approaches in reconstructing the high-resolution time series across both stable and volatile regions, ranking as the top two methods for recovering lost details at sharp peaks.

Table 11: Quantitative comparison on nine public datasets *w.r.t.* SSR. The metrics are MSE and DTW distance. For each group, the first sub-row reports the MSE and the second sub-row reports the DTW distance, both in the form of 'mean (standard deviation)' calculated over ten runs. All reported values are multiplied by 100 for ease of comparison. Based on the mean value for each metric, the top result is marked in bold and the second-best result is underlined.

| Method | ETTm1 | ETTm2 | ETTh1 | ETTh2 | weather | PEMS-SF | MotorImagery | SCP1 | SCP2 |
|---|---|---|---|---|---|---|---|---|---|
| *Baselines* | | | | | | | | | |
| SRDiff | 4.211 (0.327) | 1.938 (0.291) | 28.710 (3.719) | 10.912 (2.219) | 8.513 (1.677) | 13.319 (2.791) | 6.089 (0.661) | 3.108 (0.244) | 5.883 (0.327) |
| | 6.945 (0.228) | 5.704 (0.317) | 21.801 (2.710) | 11.524 (2.126) | 10.125 (1.103) | 14.767 (2.114) | 7.137 (1.050) | 9.595 (0.377) | 9.296 (0.419) |
| ResShift | 4.021 (0.400) | 2.019 (0.236) | 22.901 (3.731) | 11.006 (2.517) | 8.107 (0.591) | 18.572 (1.076) | 7.041 (0.398) | 2.833 (0.320) | 5.104 (0.241) |
| | 7.093 (0.236) | 5.891 (0.274) | 20.135 (2.915) | 11.689 (2.004) | 8.899 (1.016) | 16.033 (2.705) | 7.783 (0.608) | 9.589 (0.406) | 9.001 (0.239) |
| IDM | 3.627 (0.114) | 1.533 (0.107) | 19.620 (3.899) | 9.399 (1.110) | 3.910 (0.703) | 10.773 (2.041) | 5.880 (0.547) | 1.920 (0.317) | 4.944 (0.847) |
| | 6.441 (0.128) | 4.709 (0.124) | 17.203 (2.575) | 10.867 (1.377) | 4.511 (0.368) | 7.157 (1.083) | 7.325 (1.406) | 7.449 (0.283) | 8.962 (0.994) |
| FlowIE | 3.889 (0.355) | 1.704 (0.219) | 23.337 (5.389) | 10.110 (1.767) | 7.589 (1.204) | 14.129 (2.391) | 6.220 (0.819) | 2.226 (0.529) | 5.489 (0.671) |
| | 6.857 (0.315) | 5.513 (0.308) | 20.828 (4.719) | 10.508 (1.337) | 8.010 (1.118) | 15.225 (1.591) | 7.281 (1.179) | 7.084 (0.984) | 9.587 (1.632) |
| CSDI | 3.719 (0.129) | 1.931 (0.172) | 19.036 (3.767) | 9.893 (1.776) | 3.397 (0.208) | 10.894 (0.519) | 6.313 (0.496) | 6.601 (0.964) | 9.793 (1.015) |
| | 6.324 (0.187) | 4.418 (0.213) | 16.196 (2.017) | 10.679 (1.994) | **2.813 (0.394)** | 7.237 (0.885) | 7.266 (1.894) | 9.288 (2.051) | 10.464 (1.957) |
| FTS-Diff | 3.948 (0.395) | 1.874 (0.318) | 23.812 (4.761) | 9.598 (1.791) | 3.305 (0.791) | 11.078 (1.398) | 8.289 (2.001) | 2.717 (0.233) | 5.288 (0.549) |
| | 6.636 (0.366) | 4.441 (0.539) | 17.289 (2.079) | 10.500 (1.206) | 3.005 (0.228) | 10.537 (1.783) | 8.091 (1.881) | 7.469 (1.015) | 8.780 (1.971) |
| Diff-TS | 4.596 (0.473) | 2.119 (0.289) | 54.197 (4.397) | 11.793 (2.106) | 20.608 (2.490) | 23.294 (3.077) | 7.863 (1.138) | 4.428 (0.759) | 7.221 (0.808) |
| | 7.701 (0.561) | 5.416 (0.529) | 22.900 (4.264) | 11.413 (2.105) | 13.289 (2.591) | 15.857 (2.101) | 7.240 (1.542) | 10.200 (1.041) | 10.336 (1.895) |
| FlowTS | 3.661 (0.394) | 1.680 (0.296) | 59.401 (8.407) | 13.488 (3.124) | 10.579 (1.114) | 12.179 (2.003) | 9.377 (0.909) | 1.897 (0.305) | 7.497 (0.994) |
| | 6.673 (0.623) | 4.808 (0.803) | 23.597 (3.911) | 12.378 (2.520) | 9.311 (1.100) | 11.677 (2.018) | 7.894 (0.779) | 6.110 (0.993) | 10.866 (1.014) |
| *Ours* | | | | | | | | | |
| SRT | 2.649 (0.204) | 1.440 (0.149) | 22.505 (4.121) | 9.278 (1.756) | 3.089 (0.293) | 9.725 (1.559) | 5.592 (0.301) | 1.739 (0.183) | 4.389 (0.315) |
| | 5.703 (0.117) | **4.349 (0.546)** | 16.041 (2.336) | 10.330 (1.591) | 3.896 (0.308) | 7.070 (0.448) | 6.626 (0.894) | 6.893 (0.238) | 8.828 (1.093) |
| SRT-large | **2.539 (0.106)** | **1.417 (0.108)** | **18.809 (2.396)** | **7.611 (0.594)** | **2.821 (0.383)** | **9.606 (1.042)** | **5.138 (0.401)** | **1.622 (0.229)** | **2.943 (0.937)** |
| | **5.671 (0.089)** | 4.803 (0.128) | **16.001 (1.856)** | **10.181 (1.012)** | 3.295 (0.361) | **7.003 (1.202)** | **6.329 (1.041)** | **5.839 (0.683)** | **7.346 (0.503)** |

Table 12: Quantitative comparison on nine public datasets *w.r.t.* ASR. The metrics are MSE and DTW distance. For each group, the first sub-row reports the MSE and the second sub-row reports the DTW distance, both in the form of 'mean (standard deviation)' calculated over ten runs. All reported values are multiplied by 100 for ease of comparison. Based on the mean value for each metric, the top result is marked in bold and the second-best result is underlined.

| Method | ETTm1 | ETTm2 | ETTh1 | ETTh2 | weather | PEMS-SF | MotorImagery | SCP1 | CP2 |
|---|---|---|---|---|---|---|---|---|---|
| *Baselines* | | | | | | | | | |
| SRDiff | 4.134 (1.041) | 2.344 (0.212) | 21.637 (4.577) | 12.879 (1.486) | 4.944 (0.895) | 23.144 (2.048) | 14.421 (1.523) | 9.784 (0.553) | 11.329 (0.831) |
|  | 7.496 (1.011) | 5.589 (0.849) | 19.684 (3.228) | 13.005 (1.769) | 8.003 (1.003) | 18.749 (1.859) | 9.707 (0.884) | 9.531 (0.748) | 10.796 (0.428) |
| ResShift | 4.440 (0.767) | 2.049 (0.289) | 22.142 (5.031) | 13.440 (1.879) | 4.730 (0.858) | 22.355 (2.210) | 12.889 (1.402) | 9.703 (0.481) | 12.776 (0.829) |
|  | 7.783 (0.936) | 4.793 (0.741) | 21.803 (3.502) | 12.137 (1.684) | 7.499 (0.984) | 19.503 (1.979) | 7.539 (0.853) | 9.799 (0.684) | 13.233 (0.537) |
| IDM | 3.746 (0.401) | 1.534 (0.218) | 19.042 (2.018) | 9.707 (1.301) | 9.226 (0.759) | 12.604 (1.322) | 11.183 (1.203) | 8.544 (0.491) | 8.343 (0.761) |
|  | 6.783 (0.592) | 5.289 (0.533) | 19.833 (2.405) | 12.427 (1.481) | 9.503 (1.004) | 7.933 (1.308) | 8.393 (0.892) | 9.416 (0.849) | 10.828 (0.529) |
| FlowIE | 4.089 (0.831) | 2.110 (0.352) | 21.763 (3.710) | 12.887 (1.893) | 5.522 (0.799) | 18.884 (1.849) | 13.893 (1.426) | 9.605 (0.781) | 9.847 (0.972) |
|  | 7.337 (0.784) | 5.041 (0.431) | 19.410 (2.575) | 12.589 (2.011) | 8.427 (0.776) | 14.008 (1.638) | 9.588 (0.769) | 8.280 (0.882) | 10.609 (0.941) |
| CSDI | 4.041 (0.579) | 1.414 (0.231) | 19.668 (2.489) | 17.884 (1.593) | 16.442 (1.031) | 12.839 (1.383) | **11.041 (1.573)** | 9.239 (0.747) | 11.241 (0.857) |
|  | 7.129 (0.596) | 4.702 (0.531) | 18.330 (2.307) | 14.868 (1.429) | 7.308 (0.847) | 7.397 (1.528) | 7.599 (0.844) | 9.783 (0.703) | 11.130 (0.764) |
| FTS-Diff | 3.941 (0.410) | 1.879 (0.423) | 19.577 (2.481) | 10.739 (1.419) | 5.002 (0.693) | 17.202 (1.894) | 20.891 (1.541) | 8.228 (0.428) | 9.524 (0.889) |
|  | 6.993 (1.013) | 6.237 (0.881) | 16.139 (2.003) | 10.525 (1.395) | 8.094 (0.797) | 13.325 (1.550) | 11.230 (0.841) | 8.801 (0.836) | 9.004 (0.520) |
| Diff-TS | 4.228 (0.508) | 2.099 (0.357) | 42.110 (4.337) | 12.118 (1.439) | 10.329 (1.031) | 20.495 (1.997) | 23.294 (1.648) | 8.852 (0.639) | 9.879 (1.040) |
|  | 7.484 (0.741) | 5.978 (1.002) | 20.505 (2.408) | 12.639 (1.526) | 9.745 (0.977) | 15.855 (1.753) | 12.049 (0.683) | 8.648 (0.843) | 10.701 (0.632) |
| FlowTS | 4.131 (0.373) | 1.742 (0.386) | 46.118 (3.894) | 10.422 (1.630) | 17.565 (1.030) | 23.858 (2.005) | 24.516 (1.495) | 9.641 (0.783) | 12.874 (0.849) |
|  | 7.385 (0.847) | 5.389 (0.839) | 22.419 (2.551) | 10.903 (1.681) | 13.101 (1.052) | 20.110 (2.301) | 11.603 (0.758) | 7.838 (0.880) | 10.055 (0.738) |
| *Ours* | | | | | | | | | |
| SRT | 3.713 (0.491) | **1.401 (0.285)** | 21.114 (2.448) | 9.618 (1.425) | **3.544 (0.426)** | **12.542 (1.526)** | 13.007 (1.522) | 7.702 (0.529) | **8.004 (0.745)** |
|  | 6.922 (0.406) | **4.544 (0.781)** | **15.606 (1.785)** | **10.397 (1.322)** | 6.841 (0.539) | 7.305 (1.307) | **7.131 (0.684)** | 7.493 (0.694) | 8.510 (0.507) |
| SRT-large | **2.944 (0.381)** | 1.613 (0.309) | **18.897 (1.429)** | **8.248 (1.227)** | **2.947 (0.431)** | 13.008 (1.528) | 11.630 (1.248) | **7.206 (0.484)** | 8.198 (0.741) |
|  | **6.228 (0.353)** | 5.200 (0.823) | 17.859 (1.336) | 10.707 (1.206) | **4.533 (0.448)** | **7.180 (1.224)** | 7.203 (0.539) | **6.935 (0.577)** | **8.004 (0.639)** |

