# OpenReview forum: "SRT: Super-Resolution for Time Series via Disentangled Rectified Flow"
_ICLR.cc/2026/Conference — ICLR 2026 Poster_

### Official Review · Reviewer_RMtJ · 2025-10-28

**Soundness:** 3
**Presentation:** 2
**Contribution:** 3
**Rating:** 4
**Confidence:** 4

**Summary:**

This manuscript introduces a framework for time series super-resolution that leverages a decomposition-based approach combined with a rectified flow mechanism. The authors propose an implicit time function to align multi-resolution features along a shared temporal axis, and incorporate a velocity predictor with multi-resolution fusion to generate high-fidelity high-resolution time series. Extensive experiments are conducted across multiple datasets, demonstrating the method’s capability to achieve superior performance compared to several baseline models.

**Strengths:**

1.	The authors address an under-explored problem in signal processing i.e., time series super-resolution, by proposing a comprehensive framework that explicitly incorporates prior knowledge of the data’s structure.
2.	The idea of decomposing the time series into periodic and trend components and generating these separately is technology sound and aligns with classical signal analysis principles.
3.	Extensive experimental validation across diverse datasets and scales indicates the robustness of the proposed approach. Moreover, the ablation studies effectively highlight the importance of each component.
4.	Also, extending the model to a zero-shot setting via large-scale pretraining shows promising potential for practical deployments.

**Weaknesses:**

1.	As far as I know, rectified flow has been widely used, the key innovation of your model in time series SR filled remains unclear.
2.	The evaluation is primarily limited to controlled datasets, the model’s performance on in-the-wild images with diverse and challenging conditions remains underexplored.
3.	The computational complexity and inference speed are relatively high, which may hinder practical application, please analyze and compare the model complexity and resource consumption.
4.	There is a lack of comparison with some recent state-of-the-art SR methods, particularly those based on diffusion architectures. Besides, the ablation needs more elaboration, especially the hyperparameters setting and visualizations.

**Questions:**

Please refer to the weaknesses.

---

> ### Author Response · Authors · 2025-11-23
>
> Thank you for your assessment and all the comments as well as suggestions. We would like to respond to each of the weaknesses you mentioned and hope it will fully address your concerns.\
> \
> **Response to Weakness #1**\
> We would like to humbly emphasize the innovative aspects of the SRT model here, and we hope this can address your concerns.\
> First of all, **SRT is not a simple implementation of rectified flow**. Rather, we delve into the essentialness of time series data as well as the super-resolution task to find an elegant solution combined. Specifically, the **technical distinctions** that set SRT aside from the simple usage of rectified flow are as follows.\
> (1) The rectified flow is conditioned on temporal aligned features generated by our specifically designed Implicit Time Function (ITF). By carefully detecting on temporal dependencies, we design pattern smoothness for trend and periodic components, which respectively regularize locality and periodicity. \
> (2)  We design a novel cross-resolution attention mechanism in the velocity predictor for better usage of ITF aligned conditions. Attention to given low-resolution time steps and aggregated low resolution statistics is added for SSR and ASR, respectively.\
> (3) The architecture is designed based on a disentangled perspective, with the detail series being decomposed into periodic and trend components and we use two rectified flow to generate each of them. This design enhances not only the interpretability but the performance as well.\
> The contributions of these innovative designs are validated through the ablation study presented in **Section 4.2**.\
> \
> **Response to Weakness #2**\
> We appreciate the reviewer’s concern regarding the generalizability of our model to in-the-wild time series data. We guess by mentioning 'in-the-wild images', the intention of the reviewer is 'in-the-wild time series'.\
> Despite gathering as publicly available datasets, **the data collection process inherently reflects real-world, in-the-wild conditions**. For example, the PEMS-SF dataset comprises 15 months of continuous daily measurements collected by 963 sensors installed on freeways in the San Francisco Bay Area. These sensors monitor occupancy rates on multiple car lanes under a wide range of real-world driving, weather, and traffic conditions. Measurements were performed every 10 minutes over diverse days and times, thus naturally capturing the variability, noise, and complexity present in everyday urban traffic scenarios.\
> Although testing on publicly available datasets provides insight into the performance of SRT in real-world scenarios, **we are open to this comment and have conducted additional experiments within a limited timeframe** in order to further address the your concerns. Specifically, we utilized stock market data sampled originally at the minute level. Super-resolution for stock data is particularly challenging due to its susceptibility to influences from macroeconomy, news, regulatory policies, and even investment psychology. Given time constraints, we selected three stocks: CVX, AAPL, and IONQ. We collected data from five trading days during the regular trading session, using data from the first four days for training and the last day for testing. The SSR results with a scale factor of 5 are summarized in the following table.
>
> | data|SRDiff|ResShift|IDM|FlowIE|CSDI|FTS-Diff|Diff-TS|FlowTS|SRT|
> |:-|:-:|:-:|:-:|:-:|:-:|:-:|:-:|:-:|:-:|
> |CVX|0.104 / 0.102|0.109 / 0.093|0.103 / 0.091|0.127 / 0.114|0.096 / 0.085|0.102 / **0.077**|0.135 / 0.121|0.138 / 0.102|**0.093** / 0.082|
> |AAPL|0.125 / 0.108|0.122 / 0.116|0.126 / 0.104|0.132 / 0.108|0.119 / 0.105|0.125 / 0.102|0.133 / 0.130|0.128 / 0.115|**0.110** / **0.097**|
> |IONQ|0.109 / 0.097|0.114 / 0.095|0.107 / 0.093|0.116 / 0.090|0.109 / 0.095|0.107 / 0.100|0.114 / 0.111|0.119 / 0.109|**0.104** / **0.085**|
>
> As shown in the results in the table above, SRT is able to provide reliable results even on the more challenging stock data super-resolution task.

---

> ### Author Response · Authors · 2025-11-23
>
> **Response to Weakness #3**\
> As you pointed out, generation speed is crucial for practical applications. One of the reasons we chose rectified flow as the generative method in the SRT model is its fast sampling capability during the generation phase. Compared to diffusion-based models, rectified flow requires fewer sampling steps, thereby reducing the number of forward passes involved in the generation process. Thanks to this property, **SRT achieves high efficiency in generation compared with the baselines, second only to FlowTS. However, experimental results show that FlowTS lags significantly behind SRT in terms of TSSR performance**.\
> We have included an analysis of training and generation efficiency in **Appendix E.2 “Training and Generation Efficiency”** of the revised manuscript. If you are interested, please refer to that section for more details. As a preview, a summary table of training (per epoch in seconds) and inference speeds (per sample in seconds) across models is attached below.
>
> | stage|SRDiff|ResShift|IDM|FlowIE|CSDI|FTS-Diff|Diff-TS|FlowTS|SRT|SRT-large|
> |:-|:-:|:-:|:-:|:-:|:-:|:-:|:-:|:-:|:-:|:-:|
> |Training|2.70|3.03|3.84|3.35|1.75|2.97|1.82|1.53|1.77|-|
> |Generation|0.30|0.28|0.47|0.16|0.34|0.49|0.31|0.04|0.04|0.13|
>
> \
> **Response to Weakness #4**\
> Among the 8 baselines, SRDiff, ResShift, IDM, CSDI, FTS-Diffusion, and Diffusion-TS **do leverage diffusion architecture** for SR capability. \
> The recommendation of adding hyperparameters setting analysis is very constructive. **We follow the suggestion and add hyperparameter sensitivity analysis in Appendix E.1**, right after the introduction of hyperparameter settings in our initial submission. For the convenience of discussion, we also list key findings below.\
> (1) The batch size, initial learning rate, and the radius hyperparameter r in ITF demonstrate strong robustness with respect to model performance.\
> (2) The two kernel sizes involved in the average pooling during the decomposition of the detail sequence and the low-resolution sequence also generally exhibit strong robustness. Even with the most suboptimal combination on the PEMS-SF dataset, SRT still achieves the second-best result, only 0.001 MSE behind IDM. Furthermore, when the values of the two kernel sizes are set to be closer, the performance tends to be better.\
> \
> We sincerely hope that our responses and clarifications address your concerns and contribute positively to your reassessment of our work. Thank you once again for your thoughtful review and valuable feedback.

---

> > ### Comment · Reviewer_RMtJ · 2025-11-28
> > **Comments after rebuttal**
> >
> > Thank you for your response. After careful evaluation, I still believe that the methodological novelty of this work is somewhat limited. Using RF as a conditional prior is relatively common, and the proposed cross-resolution attention is mainly derived from existing attention mechanisms. In addition, the concerns regarding the experiments were not fully addressed, as no newly added state-of-the-art methods from 2025 were provided. Overall, I will maintain my score.

---

> > > ### Author Response · Authors · 2025-12-02
> > > **Rebuttal Part 1**
> > >
> > > Thank you very much for your feedback. \
> > > \
> > > Let us firstly address your concern w.r.t. methodological novelty. While understanding your concern regarding the novelty of individual components, we respectfully argue that the core contribution lies not in inventing every base component from scratch, but in the design of key functional modules and the novel integration and re-purposing of existing methods to solve the unique and previously under-addressed challenges of Time Series Super-Resolution (TSSR).\
> > > Our work makes a principled advance by introducing the **first** generative framework specifically designed for the nature of TSSR.\
> > > \
> > > **(1) A Novel Temporal Alignment Engine**\
> > > Instead of a generic interpolation, the ITF learns distinct continuous-time functions for trend and periodic components, explicitly encoding their unique temporal dependencies. **For the trend**, it enforces temporal smoothness, ensuring the upsampled trajectory reflects a physically plausible, slow evolution. **For the periodic component**, it explicitly captures and aligns cyclical patterns from the low-resolution context to the high-resolution timeline. This provides semantically rich and temporally coherent conditional signals, which are critical for guiding the subsequent generative process. This goes far beyond simple upsampling and is a novel contribution in itself.\
> > > \
> > > **(2) A Disentangled Generative Process**\
> > > By separating the generation of trend and periodic components, we effectively disentangle the mixed information in low-resolution inputs, allowing each specialized rectified flow to focus on its respective temporal pattern with the help of its respective conditions. This architectural inductive bias not only enhances **performance** but also significantly improves **interpretability**. The plausibility of generated trends and periodic patterns can be independently verified , making the TSSR process more transparent and trustworthy. The rectified flows are conditioned on the semantically aligned components from ITF, creating a structured generation process that moves far beyond conventional conditional generation.\
> > > \
> > > **(3) A Cross-Resolution Fusion Mechanism**\
> > > The novelty of our Cross-Resolution Attention (CRA) is its explicit design to fuse information across temporal scales. It uses the ITF aligned components as well as the low-resolution input to learn how a low-resolution context should inform the generation of high-resolution details. Its **dual-mode design** is a direct, problem-aware instantiation motivated by the fundamental SSR/ASR taxonomy.\
> > > \
> > > **(4) A Scalable and Generalizable Model Family**\
> > > By scaling SRT to SRT-large through increased model capacity and large-scale pre-training, we enable **zero-shot super-resolution capability, a feature not supported by the base model**. This expansion not only enhances performance but also improves robustness across multiple upsampling factors, demonstrating the framework's scalability and practical applicability in real-world scenarios where high-resolution training data is unavailable.\
> > > \
> > > In essence, the innovation of our work lies in the cohesive integration of a novel problem formalization, theoretically-grounded alignment and priors, and problem-aware instantiations, which collectively form a comprehensive and principled framework for time series super-resolution.

---

> ### Author Response · Authors · 2025-12-02
> **Rebuttal Part 2**
>
> This response also serves as an extended reply to weakness #4 raised in the official review.\
> First, we would like to clarify that ‘recency’ is not our sole criterion for selecting baselines. Unlike image super-resolution (ISR), which is a well-established research field, TSSR requires baselines to be adapted specifically for this task. In this process, if a model is 'recent' but its original design is considerably different from TSSR, substantial modifications are often necessary, which makes this recently proposed method suboptimal as a baseline.\
> We had conducted a wide survey of newly proposed diffusion-based methods from 2025, but few was found as qualified baselines. For example, DiT4SR [1] applies Diffusion Transformer (DiT) to real-world ISR. However, its framework requires text input to provide real-world prior knowledge. Similarly, models such as AdcSR, InvSR, and DiffIISR [2-4] rely on pretrained models. Transferring models pretrained on images to time series requires domain adaptation, whose effectiveness directly impacts TSSR performance, making them suboptimal choices for baselines.\
> Nevertheless, among diffusion-based ISR approaches introduced in 2025, we have identified DiT-SR [5] which is relatively suitable as a baseline. Additionally, we have also included PaD-TS [6], a recent diffusion-based time series generative model proposed in 2025.\
> DiT-SR employs a U-shaped DiT with isotropic design and is trained from scratch. While core designs such as the U-shape architecture and AdaFM are originally tailored to ISR, incorporating DiT-SR still provides a meaningful supplement for benchmarking DiT on the TSSR task. In our experiments, alignment detailed in Appendix D.1 is utilized to address the gap between inputs, and postprocessing is performed to ensure the channels of time series remain unchanged.\
> PaD-TS primarily addresses the issue of lacking explicit modeling of population-level properties in time series generation. It introduces the Same Diffusion Step Sampling mechanism to ensure that all samples in each mini-batch are at the same step during training, and incorporates Population Aware Training to regularize distribution shifts across different dimensions. In our experiments, we generate the residual sequence between high- and low-resolution series using PaD-TS, and add this to the low-resolution series to accomplish the TSSR task.\
> Using the experimental setting for Table 1, we incorporate DiT4SR and PaD-TS and the results are as follows.
> ||ETTm1(SSR)|weather(SSR)|PEMS-SF(SSR)|ETTm1(ASR)|weather(ASR)|PEMS-SF(ASR)|
> |:-|:-:|:-:|:-:|:-:|:-:|:-:|
> |DiT-SR|0.037/0.064|0.082/0.079|0.121/0.115|0.043/0.079|0.091/0.087|0.179/0.203|
> |PaD-TS|0.035/0.066|0.077/0.060|0.114/0.109|0.041/0.073|0.086/0.090|0.147/0.152|
> |SRT|**0.026/0.057**|**0.031/0.039**|**0.097/0.070**|**0.037/0.069**|**0.035/0.068**|**0.125/0.073**|
>
> In addition, we also evaluate the generation speed of both methods. Using the generation time and average rank described in Section 4.4, the results after including DiT4SR and PaD-TS are updated as follows.
> ||SRDiff|ResShift|IDM|FlowIE|DiT-SR|CSDI|FTS-Diff|Diff-TS|FlowTS|PaD-TS|SRT|
> |:-|:-:|:-:|:-:|:-:|:-:|:-:|:-:|:-:|:-:|:-:|:-:|
> |Time|0.30|0.28|0.47|0.16|0.39|0.34|0.49|0.31|0.04|0.33|0.04|
> |Rank|8.00|8.25|3.75|6.50|7.25|3.50|4.17|9.67|8.33|5.33|1.25|
>
> It can be observed that, due to the multi-step sampling mechanism, the two diffusion-based methods lag significantly behind those based on rectified flow (FlowIE and FlowTS) and SRT w.r.t. generation speed. Moreover, both DiT4SR and PaD-TS achieve only moderate overall TSSR performance. In summary, although these two latest diffusion-based methods exhibit strong capabilities in their respective domains, they are not able to efficiently and accurately address the TSSR task as SRT does.\
> \
> [1] Duan Z P, Zhang J, Jin X, et al. Dit4sr: Taming diffusion transformer for real-world image super-resolution[C] Proceedings of the IEEE/CVF International Conference on Computer Vision. 2025: 18948-18958.\
> [2] Chen B, Li G, Wu R, et al. Adversarial diffusion compression for real-world image super-resolution[C] Proceedings of the Computer Vision and Pattern Recognition Conference. 2025: 28208-28220.\
> [3] Yue Z, Liao K, Loy C C. Arbitrary-steps image super-resolution via diffusion inversion[C] Proceedings of the Computer Vision and Pattern Recognition Conference. 2025: 23153-23163.\
> [4] Li X, Wang Z, Zou Y, et al. Difiisr: A diffusion model with gradient guidance for infrared image super-resolution[C] Proceedings of the Computer Vision and Pattern Recognition Conference. 2025: 7534-7544.\
> [5] Cheng K, Yu L, Tu Z, et al. Effective diffusion transformer architecture for image super-resolution[C] Proceedings of the AAAI Conference on Artificial Intelligence. 2025, 39(3): 2455-2463.\
> [6] Li Y, Meng H, Bi Z, et al. Population Aware Diffusion for Time Series Generation[C] Proceedings of the AAAI Conference on Artificial Intelligence. 2025, 39(17): 18520-18529.

---

### Official Review · Reviewer_jCpF · 2025-10-30

**Soundness:** 3
**Presentation:** 3
**Contribution:** 3
**Rating:** 6
**Confidence:** 4

**Summary:**

This paper proposes SRT, a novel disentangled rectified flow framework for time series super-resolution (TSSR), along with its large-scale pre-trained variant SRT-large. The work makes a solid conceptual and methodological contribution by adapting rectified flow modeling, which was originally developed for image generation, to temporal data through a structured decomposition and alignment pipeline. The proposed framework demonstrates state-of-the-art results across nine datasets

**Strengths:**

1. Consistent improvements have been achieved over the baselines (SRDiff, IDM, FTS-Diff, FlowTS).

2. The ablation experiments are extensive and well-designed, quantifying the contributions of each module.

3. The zero-shot generalization capability of the SRT-large model is remarkable, echoing the scaling trends of foundational models in language and vision.

**Weaknesses:**

1. There are cases where formulas in the experiments lack numbering. Some formulas are complex and difficult to understand, though their principles can be roughly grasped by comparing them with the code. For instance, Formula 3, which lacks a number, could have a more detailed introduction of Vs. The variables c, l, and theta in Formula 3 and Formula 4 should also be described more thoroughly.

2. Although it is claimed that decomposition enhances interpretability, no visualizations or attribution analyses (such as how trend and seasonal components affect the final reconstruction) are presented.

3. While the empirical performance is excellent, the theoretical reasoning behind using rectified flow for continuous temporal domains (as opposed to standard diffusion) could be further elaborated upon. A brief discussion on the temporal smoothness prior induced by rectified flow would strengthen the argument.

4. There is no discussion on its training efficiency, memory usage.

**Questions:**

1. Does the research in this paper belong to the migration of studies from the field of image super-resolution?

2. It is recommended to conduct visualizations of the trend and seasonality components from the decomposition to highlight the interpretability advantages of the untangling design.

3. Could annotations be added to the content in the appendix, as it is difficult to establish connections with the main text?

4. Does the improvement in model performance rely on the decomposition assumption? In the text, you mention "low-resolution" and "high-resolution." Does the kernel size of Avgpool have an impact on this process?

5. The method presented in this paper is mostly described in the text as a combination of existing methods. Please point out the differences between it and the existing methods.

6. It is physically unreasonable to infer fine-grained components solely through average pooling without distinguishing between SSR and ASR modules in super-resolution tasks. Please provide an explanation.

---

> ### Author Response · Authors · 2025-11-23
>
> Thank you for the careful reading and the useful suggestions. We would like to respond to the questions and weaknesses you mentioned and hope your concerns will be successfully addressed.\
> \
> **Response to Question #1**\
> Research on super-resolution for time series is not as popular as its counterpart in image super-resolution, which may give the impression that this work is simply a transfer from a more well-known research area. However, as discussed in the Introduction and Related Work sections, **although Time Series Super-Resolution (TSSR) and image super-resolution appear similar at first glance, they are fundamentally different tasks**. They differ in several aspects, such as data dimensionality and the priors required for super-resolution. As a result, there is a need to design novel models specifically tailored for the TSSR problem, rather than simply migrating approaches from image super-resolution.\
> \
> **Response to Question #2 and Weakness #2**\
> Thanks for pointing out this weakness and giving the corresponding recommendation. We have added a visualized analysis in **Appendix G.3 'Interpretability Analysis'** for better illustration of how the disentangled design helps out with interpretability. In summary, the contribution can be twofold.\
> (1) Decomposition allows users to observe the output of each component, making the entire super-resolution process more transparent.\
> (2) The generation of each component enables users to assess the credibility of results based on their domain knowledge, thereby enhancing the practical utility of the model.\
> \
> **Response to Question #3**\
> Thank you for this valuable suggestion. **We have incorporated an Appendix Outline at the beginning of the appendix** to facilitate readers’ understanding and navigation. We believe this revision will significantly enhance the readability of our paper.\
> \
> **Response to Question #4**\
> Regarding the first concern, we boldly infer that the reviewer’s question may have two layers of meaning. First, the reviewer would like to know whether decomposition truly brings performance improvements to SRT. The answer is yes, and this has been verified in our ablation study. In addition, the reviewer seems to ask whether SRT would be unable to complete the super-resolution task if decomposition could not be performed. Our method adopts the time series decomposition from Autoformer, as shown in Eq.1. The periodic component resulting from the decomposition is not a strict one, but rather a form of 'pseudo-seasonality'. Moreover, Eq.1 guarantees that time series decomposition can always be performed.\
> Regarding the second concern, the kernel size of average pooling acts as a hyperparameter and affects the performance of the SRT model. In the revised manuscript, we have included a hyperparameter sensitivity analysis in **Appendix E.1**, where we examine the impact of kernel size and other hyperparameters on model performance. If you are interested in this section, please refer to the relevant chapter. For ease of discussion, we summarize the key findings as follows.\
> (1) The batch size, initial learning rate, and the radius hyperparameter r in ITF demonstrate strong robustness with respect to model performance.\
> (2) The two kernel sizes involved in the average pooling during the decomposition of the detail sequence and the low-resolution sequence also generally exhibit strong robustness. Even with the most suboptimal combination on the PEMS-SF dataset, SRT still achieves the second-best result, only 0.001 MSE behind IDM. Furthermore, when the values of the two kernel sizes are set to be closer, the performance tends to be better.\
> \
> **Response to Question #5**\
> While SRT leverages aspects of rectified flow and implicit neural representation, it fundamentally differs in the following ways.\
> (1) By carefully detecting the essential characteristics of the time series data, we introduce pattern smoothness in ITF, which regularizes locality in the trend component and periodicity in the periodic component.\
> (2)  We design and incorporate a novel cross-resolution attention mechanism in the velocity predictor, allowing for better usage of ITF aligned conditions. Attention to given low-resolution time steps and aggregated low resolution statistics is added for SSR and ASR, respectively.\
> (3) Unlike directly using rectified flow to generate TSSR results, we set the detail sequence between the two resolutions as the generation target and construct the entire model based on a disentangled architecture. This design enhances not only performance but interpretability as well.\
> The contributions of these innovative designs are validated through the ablation study presented in Section 4.2.

---

> > ### Comment · Reviewer_jCpF · 2025-11-26
> >
> > Thank you for your response. You have addressed most of my previous questions. At this stage, my main concern remains focused on Question 1.
> >
> > Based on your current answer, I feel that my core concern has not yet been fully addressed. You mentioned that the main contribution of your method lies in the improvements related to “data dimensionality and priors required for super-resolution.” However, from my perspective, if other techniques such as reinforcement learning, wavelet transforms, or few-shot adaptation were introduced into the same task, they would also require certain structural or hyperparameter adjustments. These modifications would typically be regarded as task-specific adaptation rather than representing genuine innovation in temporal-pattern modeling.
> >
> > What I really want to understand is the following:
> > When applied to time series, does your method introduce any substantive innovation in modeling temporal regularities? Are there explicit mathematical formulations, loss functions, or mechanism designs that directly capture temporal structures, trends, or periodic patterns, rather than merely enhancing the architecture?
> > While reading the paper, I did not find a clear distinction between this work and other recent super-resolution approaches that also use structural decomposition or prior-based modeling. For instance, is the trend–period decomposition mainly serving as a functional partition of the model architecture, or does it theoretically contribute to a deeper understanding and representation of temporal dependencies? In other words, is there a principled connection between this decomposition and the super-resolution task itself, beyond empirical structural design?
> >
> > If you could point out the mathematical expressions, derivations, or specific modules in the paper that directly relate to temporal-pattern modeling, it would greatly help clarify the core contribution. At the same time, please feel free to correct me if I have misunderstood any aspects of your approach.
> >
> > Overall, I believe that the paper is well-developed and shows a certain degree of originality. In my view, it deserves a score of 6. However, if the core innovation regarding temporal pattern exploitation could be presented more clearly and explicitly, it would significantly strengthen the contribution and overall impact of the work.

---

> ### Author Response · Authors · 2025-11-23
>
> **Response to Question #6 and Weakness #3**\
> It is necessary to clarify that the role of average pooling in SRT is **not** to infer fine-grained components, but simply to extract the trend component from the detail sequence. In contrast, the inference of fine-grained components is mainly achieved through ITF, which generates the high-resolution conditions, and rectified flow, which enables conditional generation of the detail components.\
> Additionally, in our initial submission, we have discussed the differences between rectified flow and diffusion in **Appendix G.1**, where the efficiency advantages of rectified flow over diffusion probalistic models for TSSR tasks are demonstrated.\
> \
> **Response to Weakness #1**\
> There are generally two conventions for numbering equations: numbering all equations, or only numbering those that are referenced later in the text. In writing our paper, we adopted the latter approach, as we believe it improves readability. Moreover, several accepted papers in related fields at ICLR in previous years have also used this convention [1, 2, 3, 4]. Additionally, we would like to point out that for the two equations mentioned in the comment, the meaning of variables have all been stated in the main text.\
> [1] Zhao B, Gower R M, Walters R, et al. Improving Convergence and Generalization Using Parameter Symmetries[C] The Twelfth International Conference on Learning Representations. 2024.\
> [2] Wu Y, Huang L K, Wang R, et al. Meta continual learning revisited: Implicitly enhancing online hessian approximation via variance reduction[C] The Twelfth international conference on learning representations. 2024.\
> [3] Han X, Absar S, Zhang L, et al. Root Cause Analysis of Anomalies in Multivariate Time Series through Granger Causal Discovery[C] The Thirteenth International Conference on Learning Representations. 2025.\
> [4] Lu C, Song Y. Simplifying, Stabilizing and Scaling Continuous-time Consistency Models[C] The Thirteenth International Conference on Learning Representations. 2025.\
> \
> **Response to Weakness #4**\
> Thanks for pointing out this issue. We have updated the manuscript to include an efficiency analysis of both the training and inference phases in **Appendix E.2 “Training and Generation Efficiency”**. If interested, please refer to the relevant section in the revised paper. For ease of discussion, we summarize the main findings below.\
> (1) SRT achieves the second fastest generation speed, closely following FlowTS. However, there is a substantial gap in super-resolution performance between FlowTS and SRT.\
> (2) While FlowIE incorporates rectified flow, its advantage in speed is not fully realized because, as an image super-resolution algorithm, it must perform upscaling in both spatial dimensions and apply postprocessing to maintain the resolution along channel dimension after input adaptation.\
> (3) Due to the channel-independent design of SRT-large, super-resolution must be performed separately on each dimension, and the increased number of parameters also leads to slightly slower generation speed compared to the standard SRT.\
> \
> Thank you for your valuable and detailed feedback. We have responded to all of your comments and have revised the paper to address the points you raised. We hope these improvements and clarifications will be helpful for your final assessment of our work.

---

> ### Author Response · Authors · 2025-11-26
>
> Thank you very much for your thoughtful feedback. We are glad that most of your questions have now been resolved.\
> Regarding Question #1, in our previous response, we mistakenly assumed that this was a relatively simple yes-or-no question. As a result, we only provided a general overview of the differences in priors and data dimensionality between time series super-resolution and image super-resolution. This may have led you to believe that the main contribution of our work lies in data dimensionality alignment and prior adaptation from one field to another. However, what we were actually emphasizing in the previous reply was the motivation for designing innovative methods, rather than the innovation itself. \
> In light of your latest comments, we would like to formally clarify and summarize the substantive contributions of our work in modeling temporal patterns, as well as the reasoning behind them.\
> \
> **1. Foundational Principle: Constraining the Ill-posed TSSR via Disentangled Dynamics**\
> The core challenge of TSSR is its severe ill-posedness. Our fundamental innovation is the introduction of a structured prior that decomposes the ambiguous high-resolution signal into two semi-independently evolving components, i.e., a slowly-varying trend and localized periodic details.\
> This is not merely an architectural partition but a principled modeling choice grounded in temporal dynamics. **It transforms an unstructured generation task into two well-regularized sub-problems, where the generation of each component is explicitly conditioned on its semantically aligned, low-resolution counterpart.** This disentanglement serves as a powerful inductive bias, drastically reducing the solution space and ensuring that the final output is not only plausible but also composed of verifiable, physically meaningful parts.\
> \
> **2. A Cohesive Framework of Temporal Modeling Innovations**\
> This core principle is realized through several detailed innovations.\
> **(2.1) Temporal Enrichment.** While the implementation in **Eq.2** may seem simple, it fundamentally expresses a critical difference between time series and image modeling. Temporal dependencies in time series often span much longer timescales, in contrast to image modeling, which typically relies on local pixel neighborhoods. Therefore, we use a larger receptive field to expand the information available at each time step, providing the implicit neural representation with richer temporal context for modeling.\
> **(2.2) Pattern Smoothness.** This component ensures that, for any time step on the target time axis, its mapped value on the time series aligns with temporal dependencies. For the trend component, values on the target time axis are constrained to follow local temporal correlations, as formalized in **Eq.4**. This equation ensures that the closer a coordinate on the target time axis is to a point on the original time axis, the greater its influence. Eq.4 effectively prevents unreasonable jumps or discontinuities in the recovered trend that would contradict the low-resolution pattern’s underlying smoothness. For the periodic component, we require that values on the target time axis reflect periodicities present in the original time axis. Thus, in **Eq.5**, we assign greater weights to original time steps that are spaced by one period, enabling authentic low-frequency periodic structures to be represented in the target axis.\
> **(2.3) Cross-Resolution Temporal Dependency.** The Cross-Resolution Attention (CRA) is explicitly designed to model dependencies across different temporal scales. It is not a standard attention mechanism but one that learns to map a low-resolution context to a high-resolution pattern. The **gating mechanism for SSR** enables the model to correct errors at observed low-resolution time points, while **broadcasting for ASR** distributes low-resolution information and facilitates global information learning. SSR and ASR are two unique sub-tasks in time series super-resolution, and CRA offers an effective and concise solution, establishing a learnable bridge between different resolutions.\
> \
> In summary, the disentangled dynamics (w.r.t. 1) is the central thesis, while the ITF (w.r.t. 2.1 and 2.2) and CRA (w.r.t. 2.3) are its key enablers, together forming a principled and cohesive framework for temporal super-resolution. We believe this clearly distinguishes our work as an innovation in temporal-pattern modeling, beyond task-specific adaptation.

---

> > ### Comment · Reviewer_jCpF · 2025-11-27
> >
> > Thank you for your response. The core innovation of the paper appears to be largely a combination of Autoformer-style decomposition and rectified flow techniques from super resolution. Based on both your reply and my reading of the paper, the way you address the problem mainly builds upon existing methods instead of introducing new theoretical insights, which is the primary reason for my score of 6.

---

> > > ### Author Response · Authors · 2025-11-30
> > >
> > > Thank you for the follow-up comment. We understand your perspective that some of SRT's components, like decomposition and rectified flow, have precedents. **However, we respectfully argue that the primary innovation lies not in the ingredients themselves, but in the novel recipe we created to solve a fundamentally reconceptualized problem.**\
> > > \
> > > **Our first key contribution is a conceptual reorganization of the TSSR landscape.** We identified and formally defined two distinct problem types based on low-resolution data genesis, i.e.,  Sampled (SSR) and Aggregated (ASR) Super-Resolution. This taxonomy is critical because SSR (generating missing samples) and ASR (distributing an aggregate value) present different mathematical challenges, which had been conflated in prior work and demand tailored solutions.\
> > > This new problem framing directly motivated and justified our technical synthesis.\
> > > **(1) Decomposition Repurposed as a Generative Prior.** Prior works like Autoformer use decomposition for forecasting, where the goal is extrapolation. In contrast, we introduce a principled, decomposition-conditioned generative process for TSSR. This explicit modeling of semi-independent temporal dynamics is a novel application of decomposition for generation, not extrapolation.\
> > > **(2) Rectified Flow as a Synergistic Engine.** We selected rectified flow not merely as "a generative model from images," but for its specific property of learning straight trajectories. This property synergizes perfectly with our disentangled design, enabling efficient and stable modeling of two distinct generation paths (trend and periodicity) that would be entangled in a single, complex diffusion process.\
> > > **(3) Problem-Aware Instantiation.** The SSR/ASR distinction directly inspired technical innovations like our Cross-Resolution Attention (CRA), which uses distinct mechanisms to handle the precise constraints of each task.\
> > > \
> > > In summary, while some of the ingredients are known, **a cohesive framework from problem taxonomy to theoretically-grounded prior (disentangled generation) to synergistic model choice is entirely novel.** This end-to-end rethinking constitutes our core contribution to time-series modeling.

---

### Official Review · Reviewer_dVw6 · 2025-10-30

**Soundness:** 3
**Presentation:** 2
**Contribution:** 3
**Rating:** 4
**Confidence:** 3

**Summary:**

The paper presents Super-Resolution for Time Series (SRT), a method that reconstructs high-resolution time series from low-resolution data. SRT decomposes the input into trend and seasonal components, using rectified flow and cross-resolution attention to generate details. An extended version, SRT-large, offers zero-shot capabilities with large-scale pretraining. Experiments show that SRT outperforms existing methods in accuracy and robustness across multiple datasets and tasks.

**Strengths:**

The paper presents a fresh approach to time series super-resolution, combining time series decomposition, implicit time functions, and rectified flow. The proposed model is carefully designed, with a clear explanation of its architecture, including the innovative cross-resolution attention mechanism.

**Weaknesses:**

While SRT outperforms other methods on the tested datasets, it’s unclear how well the model generalizes to other, unseen datasets. Future experiments on more diverse datasets might be necessary to confirm whether the high performance is consistent across varied domains. Including more methods, especially more recent advancements or other domain-specific models, could provide a clearer picture of SRT's performance in a broader context. The paper includes a quantitative comparison but doesn’t showcase qualitative results

**Questions:**

1.	Could the authors include visual examples of the reconstructed high-resolution time series from both the SRT model and the baseline methods?
2.	The Implicit Time Function (ITF) seems crucial to the model's performance. How does it perform in other types of time series tasks, like forecasting or anomaly detection? Would it be effective for multi-modal time series data?
3.	Due to the complex generative mechanisms introduced by the SRT model (such as the implicit time function and velocity predictor), is the computational complexity during the inference stage reasonable? It would be helpful to include a comparison of computational complexity or inference time.

---

> ### Author Response · Authors · 2025-11-23
>
> Thanks for your time of thoroughly reading our work and giving constructive suggestions. Below, please find the responses to some specific comments and we hope your concerns will be addressed successfully.\
> \
> **Response to Question #1**\
> Visualized examples of reconstructed high-resolution time series are significant in demonstrating SRT's leading TSSR performance compared with other baselines.\
> **However, we have to clarify that in our initial submission, visualized results are included in Appendix H, and citation to the Appendix is made at the beginning of Section 4.1**.\
> \
> **Response to Question #2**\
> The Implicit Time Function (ITF) does perform a crucial role in our proposed SRT model. It generates temporal aligned conditions for later velocity prediction when implementing rectified flow. As for this specific question, since **both forecasting and anomaly detection are essentially different tasks compared with time series super-resolution**, ITF's effectiveness in these two tasks has not been discussed in our submission.\
> **However, we are open to this question and conduct a brief experiment to find the answer.**\
> Due to the limited time available during the discussion phase, we only conducted a brief experiment on the forecasting task. We use public datasets ETTm1 and ETTm2 with OT as forecasting target, and downsample MUFL and MULL to hourly to create multi-resolution covariates. We choose DeepAR, Informer, and PatchTST as forecasters. With forecast horizon as 32 and 96, and the corresponding lookback period as 192 and 576, the performance in MSE between direct forecasting and ITF covariates-aligned forecasting is demonstrated in the table below.
>
> | datasets|predictors|direct forecast|ITF-aligned|improvement|direct forecast|ITF-aligned|improvement|
> |:-|:-|:-:|:-:|:-:|:-:|:-:|:-:|
> |ETTm1|DeepAR|0.0157|0.0152|0.0005|0.0904|0.0826|0.0078|
> |ETTm1|Informer|0.0220|0.0209|0.0011|0.0901|0.0930|(0.0029)|
> |ETTm1|PatchTST|0.0146|0.0155|(0.0009)|0.0697|0.0675|0.0022|
> |ETTm2|DeepAR|0.0184|0.0177|0.0007|0.1164|0.1201|(0.0037)|
> |ETTm2|Informer|0.1110|0.0975|0.0135|0.2668|0.1873|0.0795|
> |ETTm2|PatchTST|0.1004|0.0982|0.0022|0.1242|0.1044|0.0198|
>
> We found that employing ITF generally had a positive impact on prediction accuracy in most scenarios, but also resulted in a negative effect in a few cases. This may be because, in the original design of ITF, its purpose was to generate high-resolution conditions from low-resolution periodic and trend components, which would then jointly participate in the rectified flow generation of detail sequences. In this context, ITF and the SRT model both operate on the same dimension of the time series. However, in the forecasting setting, the role of ITF is simply to align covariates in the time dimension.\
> We hope the supplementary experiments will address your question. And since it has no direct correlation with the main idea of this paper, the above experimental results will not be added to the main paper.\
> \
> **Response to Question #3**\
> Thank you for this question, and we are pleased to address your concern.\
> SRT employs rectified flow as its generative model, and one of its key advantages is fast sampling during generation. In SRT, we use a four-step generation process to transfer two randomly initialized noise vectors to the trend and periodic components, thereby accomplishing the TSSR task. Although the theoretical foundation of ITF is relatively complex, the network for value prediction **adopts a simple structure**. Moreover, the velocity predictor is involved in **only four forward passes** during generation, so the overall generation remains efficient.\
> A full analysis and the corresponding results are provided in **Appendix E.2 "Training and Generation Efficiency"**. We invite you to consult this section for a comprehensive overview of the results.\
> \
> We hope that our clarifications and responses have addressed your concerns. We sincerely appreciate the time you have devoted to reviewing our paper, as well as the valuable questions and suggestions you have provided. We would be grateful if you could consider reevaluating your review in light of our responses.

---

> > ### Comment · Reviewer_dVw6 · 2025-11-26
> >
> > Thanks for your detailed rebuttal.
> >
> > The authors have partially addressed my questions. However, despite the inclusion of time comparisons in Table 6, the paper still lacks a systematic analysis of computational complexity—such as FLOPs or step-wise inference overhead—which makes it difficult to precisely assess the cost introduced by the 4-step generation mechanism and the omission of the reflow operation. In addition, while the training time of SRT-large is reported, the paper does not further discuss the trade-off between its significantly larger parameter scale and the corresponding performance gains, nor does it evaluate the inference-time resource requirements in practical deployment scenarios. A more detailed analysis would help clarify the efficiency and applicability of both models.

---

> ### Author Response · Authors · 2025-11-27
>
> We sincerely thank the reviewer for the constructive feedback and for prompting us to provide a more systematic analysis. We appreciate the opportunity to clarify our approach and have now supplemented our efficiency evaluation with a comprehensive computational complexity analysis.\
> \
> **FLOPs Analysis**\
> In our previous response, we focused on empirical inference time mainly because it encompasses the entire generation pipeline, including system-level overhead (e.g., data loading and communication), whereas FLOPs only quantify computational operations. Although reporting FLOPs is more favorable for our lightweight SRT model, we still use inference time in our previous response, as we believe this holistic view is essential for evaluating overall efficiency in practical tasks.\
> However, we fully acknowledge the reviewer's point that FLOPs provide a standardized, hardware-independent perspective on model complexity, and we have now addressed this gap.\
> As requested, we have computed the FLOPs for all models with input to be super-resolved at the size of [1, 337, 7]. The calculations cover the full generation process, including multi-step sampling as well as any relevant pre-processing and post-processing operations. The results are summarized in the table below.
>
> | models|SRDiff|ResShift|IDM|FlowIE|CSDI|FTS-Diff|Diff-TS|FlowTS|SRT|SRT-large|
> |:-|:-:|:-:|:-:|:-:|:-:|:-:|:-:|:-:|:-:|:-:|
> |GFLOPs|251.01|204.17|317.16|14.39|230.63|288.03|187.49|3.74|2.49|35.46|
>
> The FLOPs analysis confirms that SRT achieves a favorable balance between efficiency and performance. Its 4-step-sampling mechanism introduces least computational overhead. Comparison between all the baseline models reveals that the overall trend is that rectified flow based methods conduct generation with significantly lower FLOPs than diffusion-based methods. This finding also supports our choice of the rectified flow framework for its superior computational efficiency.\
> \
> **Trade-off between two SRT versions in practice**\
> The primary value of SRT-large is twofold. Firstly, it pushes the performance ceiling of supervised TSSR. Secondly, it unlocks strong zero-shot super-resolution capability, a feature the standard SRT does not possess. \
> This enhanced capability comes with predictable costs in inference efficiency, which we have now quantified. In addition to the FLOPs and inference time reported previously, we have measured the peak GPU memory consumption. Furthermore, to assess practical deployability, we also evaluated both models on a V100 GPU, a more common and less powerful chip than the A100. Specifically, for the same [1, 337, 7] size pseudo input, the peak GPU memory consumption for SRT is 0.03 GB, compared with SRT-large's 0.26 GB. This substantial increase explains why SRT-large necessitates a smaller batch size (e.g., a maximum of 8 on big datasets like MotorImagery) on V100 GPU, which directly impacts the total processing time for large data volumes.\
> In summary, the choice between SRT and SRT-large presents a clear trade-off for practitioners.\
> (1) Choose SRT for scenarios where historical high-resolution data is available and low latency is a clear requirement.\
> (2) Choose SRT-large when zero-shot capability is critical and sufficient computational resources are available.\
> \
> We have incorporated the above analysis into the paper (at the end of **Section 5** for trade-off analysis and **Appendix E.2** for FLOPs analysis) to better guide potential users and clarify the applicability of each model variant.

---

### Official Review · Reviewer_NJg4 · 2025-10-31

**Soundness:** 3
**Presentation:** 3
**Contribution:** 3
**Rating:** 8
**Confidence:** 2

**Summary:**

This paper introduces SRT, a novel framework for time series super-resolution (TSSR), which aims to reconstruct high-resolution signals from low-resolution inputs. The authors formally distinguish two TSSR subtypes: Sampled Super-Resolution (SSR) and the more challenging Aggregated Super-Resolution (ASR). The core of SRT involves disentangling the target high-resolution details into trend and seasonal components, generating them in parallel via separate rectified flows. Key innovations include an Implicit Time Function (ITF) for continuous temporal alignment and a Cross-Resolution Attention (CRA) mechanism within a decoder-only velocity predictor to fuse multi-resolution information. The authors also present SRT-large, a scaled-up, pre-trained model that demonstrates strong zero-shot TSSR capability. Extensive experiments on nine datasets show that SRT outperforms adapted baselines from image SR and time series generation.

**Strengths:**

- **Originality:** The work makes several original contributions. It is among the first to formally define and tackle the distinct problems of SSR and ASR in time series. The proposed architecture is a creative and non-trivial synthesis of time series decomposition, implicit neural representations, and modern generative models (rectified flow), moving beyond simple adaptations of image-based methods.

- **Significance:** The ability to perform high-fidelity TSSR, especially for the ill-posed ASR task and in a zero-shot setting, is highly significant for many real-world applications where high-resolution data collection is constrained. The proposed benchmark and clear problem definitions provide a solid foundation for future research.

- **Quality and Clarity:** The paper is well-structured and the method is clearly explained. The experimental design is thorough, evaluating on nine diverse datasets and using both point-wise (MSE) and structural (DTW) metrics. The comprehensive ablation studies and component analysis convincingly validate the design choices. The introduction of a large-scale pre-trained model for zero-shot super-resolution is a notable contribution.

**Weaknesses:**

- **Computational Efficiency:** While the rectified flow enables faster sampling than diffusion models, the paper does not discuss the overall training or inference cost of SRT and SRT-large. The computational burden of the two-stage training (PD and then reverse diffusion) and the large-scale pre-training for SRT-large could be a practical limitation. A comparison of inference time with baselines would be informative.

- **Limitation of Decomposition Priors:** The method heavily relies on the assumption that time series can be effectively disentangled into trend and seasonal components. While this holds for many periodic signals, its performance on time series with irregular, non-stationary, or abrupt transient patterns is less clear and could be a limitation.

- **Baseline Adaptations:** The baselines are primarily adapted from other domains (image SR) or tasks (imputation/generation). While this is understandable given the nascent stage of TSSR, it would strengthen the paper to include comparisons with a broader range of time-series-specific interpolation or reconstruction techniques, even if simpler, to better contextualize the performance gain.

**Questions:**

see weaknesses

---

> ### Author Response · Authors · 2025-11-23
>
> Thanks for your careful reading and the appreciation of our work. Here we would like to respond to each of the weaknesses you mentioned. \
> \
> **Response to Weakness #1** \
> Thanks for pointing out this issue. \
> An important consideration in the design of the SRT model is the fast sampling property of rectified flow. To further demonstrate the high efficiency of SRT on TSSR task, we have added **Appendix E.2 “Training and Generation Efficiency”** to the revised manuscript, where we analyze the time consumption of SRT and baseline models in both the training and generation phases. Please refer to the corresponding section for details. Here, we summarize the key findings as follows. \
> (1) SRT achieves the second fastest generation speed, closely following FlowTS. However, there is a substantial gap in super-resolution performance between FlowTS and SRT.\
> (2) While FlowIE incorporates rectified flow, its advantage in speed is not fully realized because, as an image super-resolution algorithm, it must perform upscaling in both spatial dimensions and apply postprocessing to maintain the resolution along channel dimension after input adaptation.\
> (3) Due to the channel-independent design of SRT-large, super-resolution must be performed separately on each dimension, and the increased number of parameters also leads to slower generation speed compared to the standard SRT.\
> \
> **Response to Weakness #2**\
> Thank you for pointing out the potential limitation regarding the decomposition prior. As with image super-resolution, time-series super-resolution is an inherently ill-posed problem, requiring incorporation of priors to make the problem tractable and meaningful. As you mentioned, one of the priors we chose is time series disentanglement, which is widely adopted and validated from supervised learning (e.g. Autoformer) to self-supervised learning (e.g. CoST). Specifically, we chose Autoformer's decomposition method, which is capable of decomposition whether the time series demonstrate strong periodicity or not. This is valid on datasets like MotorImagery whose detail sequence does not demonstrate significant periodicity, where SRT still delivers the leading performance.\
> Recognizing that no single prior can fit all cases, we have discussed this limitation and our future work explicitly in Section 6 of our submission. \
> \
> **Response to Weakness #3** \
> Thank you for your insightful suggestion. We acknowledge that our current baselines are primarily adapted from image SR and general imputation/generation tasks, due to the lack of established benchmarks in the nascent field of TSSR. Nonetheless, we agree that comparing against a broader range of time series-specific interpolation techniques can help to better contextualize the performance gains of our method.\
> We have included comparisons with three classic approaches, which are nearest-neighbor (NN), linear interpolation (Linear), and cubic spline interpolation (Spline). These baselines provide a direct reference for the original domain and help highlight the advantages of our method in terms of TSSR accuracy and temporal consistency. Using the same shuffle, the results on ETTm1, weather, and PEMS-SF are demonstrated in the table below.
>
> | Methods |ETTm1(SSR)|weather(SSR)|PEMS-SF(SSR)|ETTm1(ASR)|weather(ASR)|PEMS-SF(ASR)|
> |:-|:-:|:-:|:-:|:-:|:-:|:-:|
> |NN|0.070 / 0.059|0.066 / 0.056|0.138 / 0.093|0.053 / 0.084|0.059 / 0.080|0.150 / 0.119|
> |Linear|0.046 / 0.065|0.051 / 0.041|0.099 / 0.087|0.053 / 0.079|0.063 / 0.085|0.146 / 0.096|
> |Spline|0.051 / 0.064|0.059 / **0.037**|0.101 / 0.089|0.052 / 0.075|0.062 / 0.091|0.146 / 0.093|
> |SRT|**0.026** / **0.057**|**0.031** / 0.039|**0.097** / **0.070**|**0.037** / **0.069**|**0.035** / **0.068**|**0.125** / **0.073**|
>
> In the revised version of the manuscript, we have included the aforementioned **quantitative** results as well as the corresponding **visualizations** in **Appendix G.2, "Comparison with Classic Time Series Interpolation Methods"**. The visualizations reveal that nearest neighbor interpolation and linear interpolation exhibit significant local deviations from the high-resolution ground truth, resulting in distortions in TSSR results. Although cubic spline interpolation is better able to capture the overall trend of the time series, its inherently polynomial fitting of the observed low-resolution data points leads to over-smoothing and, consequently, a loss of sharp details around the peak.\
> \
> Thank you very much for your positive evaluation and thoughtful comments. We sincerely hope that our detailed responses have addressed the limitations you highlighted, and can further reinforce your confidence in the strengths and contributions of our work.

---

> ### Author Response · Authors · 2025-11-28
>
> Dear Reviewer,\
> Thank you for your time and feedback on our paper. We noticed that after our rebuttal,  no additional comments have been received. We are keen to ensure that our responses adequately addressed your concerns, and we would greatly appreciate any additional feedback you might have.\
> If there are any aspects of our work or rebuttal that remain unclear, we would be happy to provide further clarification or additional details. Our goal is to ensure that you have all the information needed to assess our paper confidently.\
> Thank you again for your consideration, and we look forward to your response.\
> Sincerely\
> Authors
>
> **Update:**\
> Additionally, we have newly uploaded a new revision of our paper. All significant changes compared to the initial submission are listed below:\
> (1) Addition of Section 4.4, which compares generation efficiency across all models.\
> (2) Inclusion of GPU memory requirements and trade-off analysis between the two versions of SRT in Section 5.\
> (3) Addition of an appendix outline at the beginning of the appendix to facilitate navigation.\
> (4) Hyperparameter sensitivity analysis has been incorporated into Appendix E.1.\
> (5) Appendix E.2 now discusses training and generation speed, memory requirements, and practical deployment considerations.\
> (6) Inclusion of Appendix G.2, which compares SRT with three classic time series interpolation methods.\
> (7) Addition of Appendix G.3, detailing SRT’s interpretability in TSSR and its usability in practical scenarios.

---

### Author Response · Authors · 2025-12-03
**Summary to Assist Decision**

Dear Area Chair,\
\
 &nbsp; &nbsp; &nbsp; &nbsp; Thank you for your time and effort in overseeing the review process of our paper. We are aware of the increased responsibilities involved in the decision-making process this year. Below, we provide a summary that highlights the contributions of our work and key responses to reviewers’ comments, hoping it will aid in your final assessment.\
\
**Contribution Summary**\
 &nbsp; &nbsp; &nbsp; &nbsp; Our work addresses a previously underexplored yet practical problem, i.e., Time Series Super-Resolution (TSSR). We systematically formulate TSSR into two distinct sub-tasks, Sampled (SSR) and Aggregated Super-Resolution (ASR), and propose SRT, a method that employs two disentangled rectified flows to conditionally generate high-resolution periodicity and trend components respectively. We design the ITF to achieve temporal alignment for different types of temporal components, and the CRA mechanism to effectively fuse information between resolutions. Furthermore, to tackle certain TSSR scenarios where no high-resolution label is available, we scale SRT to SRT-large, equipping the model with zero-shot capability. As one of the pioneering systematic studies in this area, our paper offers a clear problem formulation, a principled framework, and extensive experimental validation.\
\
**Discussion Summary**\
 &nbsp; &nbsp; &nbsp; &nbsp; Overall, the reviewers recognized the contributions of our work. All four reviewers provided positive feedback on the **novelty** of the problem and solution (e.g., 'The paper presents a fresh approach ...' by dVw6, and 'The proposed architecture is a creative and non-trivial synthesis of ...' by NJg4), as well as the **performance** of SRT (e.g., 'Consistent improvements have been achieved ...' by jCpF, and '... indicates the robustness of the proposed approach' by RMtJ). Reviewer jCpF and RMtJ commended the **zero-shot capability** of SRT-large by commenting 'the zero-shot generalization capability of the SRT-large model is remarkable' and 'extending the model to a zero-shot setting via large-scale pretraining shows promising potential for practical deployments', respectively. In terms of **clarity**, Reviewer NJg4 praised that 'the paper is well-structured and the method is clearly explained' and Reviewer dVw6 credited the 'clear explanation' of our paper.\
 &nbsp; &nbsp; &nbsp; &nbsp; For constructive suggestions given by reviewers, such as those concerning efficiency analysis, hyperparameter sensitivity analysis, and visualized analysis of interpretability, we have actively addressed them not only in our rebuttal but also by updating the corresponding sections in the manuscript.\
 &nbsp; &nbsp; &nbsp; &nbsp; For some requests that were less directly aligned with the core problem, we maintained open-minded and responded within the limited time. For example, Reviewer dVw6 inquired about the utility of ITF for forecasting and anomaly detection, which are two totally distinct tasks from TSSR, yet we designed and conducted additional experiments to explore this potential ITF utilization. Additionally, Reviewer RMtJ appeared to conflate image super-resolution with TSSR, suggesting experiments with ‘in-the-wild images’ and requesting comparisons against 2025 diffusion-based SR methods. We have clarified this distinction and, where feasible, conducted relevant supplementary experiments.\
 &nbsp; &nbsp; &nbsp; &nbsp; By the close of the discussion period, most concerns from the reviewers had been resolved. For follow-up requests, such as Reviewer RMtJ’s regarding newer baselines and Reviewer dVw6’s on FLOPs analysis, we have since performed additional experiments and provided further clarification.\
\
 &nbsp; &nbsp; &nbsp; &nbsp; We sincerely thank you for your time and service in managing this process. We believe our work offers a novel and systematic contribution to time series super-resolution, and we respectfully submit it for your final consideration.\
\
Best regards,\
Authors

---

### Meta-Review · Area_Chair_gs4d · 2026-01-02

**Summary:**

This paper introduces SRT, a novel disentangled rectified flow framework for time series super-resolution, addressing the under-explored problem of reconstructing high-resolution temporal signals from low-resolution inputs. The work makes several contributions: a formal distinction between Sampled Super-Resolution and Aggregated Super-Resolution, a disentangled architecture using separate rectified flows for trend and seasonal components, an Implicit Time Function for continuous temporal alignment, a Cross-Resolution Attention mechanism, and SRT-large for zero-shot capability.

Reviewers recognized the paper's strengths, including its novelty in addressing TSSR as a distinct problem from image super-resolution, comprehensive experimental validation across nine datasets, strong performance improvements over baselines, and the promising zero-shot capability of SRT-large.

However, concerns were raised regarding methodological novelty, computational efficiency analysis, generalization to diverse real-world datasets, comparison with more recent state-of-the-art methods, and the clarity of mathematical formulations and interpretability analysis.

**Reviewer Concerns:**

The authors have provided compelling arguments about their contributions: novel problem formulation, principled integration of decomposition as a generative prior, and the ITF/CRA mechanisms specifically designed for temporal alignment. While some reviewers remain skeptical about the novelty of individual components, the comprehensive framework and empirical results represent advancement in the TSSR field.

**Reviewer Scores:**

Based on the discussion and rebuttal quality:

Reviewer NJg4 ould likely maintain a positive score given their initial support and the authors' thorough responses to their efficiency and baseline concerns. Their lowered confidence reflects unfamiliarity with the area rather than issues with the work.

Reviewer dVw6 would likely increase to 5-6 given the comprehensive FLOPs analysis, efficiency trade-off discussion, and additional experiments addressing their computational complexity concerns.

Reviewer jCpF would likely maintain 6 or possibly increase to 7 given the authors' detailed clarification of temporal pattern modeling innovations and interpretability enhancement.

Reviewer RMtJ would likely increase to 5 given the additional experiments, efficiency analysis, and hyperparameter studies.

---

### Decision · Program_Chairs · 2026-01-26

Accept (Poster)